# Adaptive Gait Generation for Hexapod Robots Based on Reinforcement Learning and Hierarchical Framework

**Zhiying Qiu [1], Wu Wei [1,2,*] and Xiongding Liu [1]**

1. School of Automation Science and Engineering, South China University of Technology, Guangzhou 510641, China
2. The Key Laboratory of Autonomous Systems and Networked Control, Ministry of Education, Unmanned Aerial Vehicle Systems Engineering Technology Research Center of Guangdong, South China University of Technology, Guangzhou 510641, China
* Correspondence: weiwu@scut.edu.cn; Tel.: +86-139-2601-3970

**Abstract:** Gait plays a decisive role in the performance of hexapod robot walking; this paper focuses on adaptive gait generation with reinforcement learning for a hexapod robot. Moreover, the hexapod robot has a high-dimensional action space and therefore it is a great challenge to use reinforcement learning to directly train the robot's joint angles. As a result, a hierarchical and modular framework and learning details are proposed in this paper, using only seven-dimensional vectors to denote the agent actions. In addition, we conduct experiments and deploy the proposed framework using a real hexapod robot. The experimental results show that superior reinforcement learning algorithms can converge in our framework, such as SAC, PPO, DDPG and TD3. Specifically, the gait policy trained in our framework can generate new adaptive hexapod gait on flat terrain, which is stable and has lower transportation cost than rhythmic gaits.

**Keywords:** hexapod robot; reinforcement learning; hierarchical framework; gait generation

## 1. Introduction

Compared with wheeled robots, a hexapod robot has many advantages, including rich gait [1], strong load capacity [2] and discrete support dependence [3]. With superior terrain adaptability and excellent performances, hexapod robots have been widely applied to disaster rescue [4], factory automation [5], intelligent maintenance [6] and exploration [7]. Nevertheless, most research still focuses on hexapod robot gait planning and motion control [8]. Regarding robotic control, hexapod robots are high-dimensional, omnidirectional and non-smooth systems with complex kinematic structures, uncertain dynamics and inherently diverse physical constraints [9,10]. In terms of robotic motion, a hexapod robot relies on the alternate support and swing of each limb to advance its body's motion [11], and therefore the movement of the hexapod robot is constrained by gait.

Locomotion in legged animals is characterized as a rhythmic behavior [12]. To achieve a rhythmic gait, some existing studies have used Central Pattern Generators (CPGs) as open-loop oscillators [13], which rely on centrally generated rhythms to drive overall behavior. Support for such approaches has been found, in particular, on fast walking in insects, such as rapid locomotion in cockroaches [14] or the running and swimming behavior of the salamander [15]. Importantly, with a high number of joints, this becomes a challenging problem, and open-loop oscillator solutions are usually not applicable [16]. This and has led to biologically inspired control approaches through combined areas of research. As one example, the Walknet approach for hexapod robots realizes a decentralized and modular structure that reflects insights from walking in stick insects [17]. While this approach can deal with a variety of disturbances during locomotion, it is still limited when dealing with novel and challenging walking situations [16]. In recent years, data-driven-based methods, such as reinforcement learning and neural networks, have attracted significant attention

because of their ability to create more accurate and robust control policies, which provides new feasible schemes for hexapod robot motion control [18].

Reinforcement learning (RL) can learn feasible planning and control strategies from data and trials. RL has been extensively studied and applied in terms of legged robot walking tasks. For instance, Peng et al. proposed a deep learning optimization strategy to train a biped robot in simulated animations, enabling the robot to pass through random obstacles freely [19]. In addition, Tan used proximal policy optimization to train a quadruped robot motion strategy and realized the transformation from simulation to physical robot [20]. Similarly, Tsounis constructed the Markov decision process (MDP) and planned trajectories in a high-dimensional, continuous, state-action space, realizing the stable movement of the robot in different environments [21]. In relevant research, Deep Reinforcement Learning (DRL) has proven itself capable of automatically acquiring control policies to accomplish a large variety of challenging locomotion tasks. Fu et al. designed a novel DRL method to implement multi-contact motion planning for hexapod robots moving on uneven plum-blossom piles [22]. Thor and Manoonpong proposed a simple yet versatile modular neural control framework with fast learning in which behavior-specific control modules could be added incrementally to obtain increasingly complex emergent locomotion behaviors [23]. Showing the impressive performance of DRL for special tasks in footed robots, Miki employed a model-free reinforcement learning approach to train a deep policy which enabled the ANYmal robot to balance a light-weight ball robustly using its limbs without any contact measurement sensor [24].

As a series-parallel composite omnidirectional mobile vehicle, the hexapod robot is a complex coupling agent. As a result, using DRL to train an end-to-end controller leads to non-convergence and gait disorder during training [25]. This has led to a growing interest in hierarchical control frameworks and how these frameworks could be exploited to improve behaviors in DRL. For example, Merel et al. proposed several such bio-inspired principles of hierarchical control and also advocated their implementation into robot architectures [26]. As one example, they emphasize a hierarchical framework with a higher-level planner modulating a lower-level controller. In addition, Eppe et al. provided the cognitive foundations of hierarchical problem-solving and proposed steps to integrate biologically inspired hierarchical mechanisms to enable advanced problem-solving skills in artificial agents [27]. Both works above provide a detailed and excellent overview for hierarchical reinforcement learning, which was significant to our paper. Similarly, Azayev and Zimmerman trained policies independently in individual scenarios using DRL, and presented a scalable two-level hierarchy for hexapod locomotion through complex terrain without the use of exteroceptive sensors [1].

This paper studies the gait generation of hexapod robots and realizes adaptive gait generation using RL and hierarchical framework. The main contributions of this paper are as follows. First, we developed a RL-based hierarchical control framework to reduce the dimensionality of the action space, which was then deployed in a real robot. Specifically, for the speed-adaptive gait generation task, we describe the whole learning process, including the MDP formulation, detailed training settings and policy gradient algorithm. Furthermore, we designed simulations and experiments to demonstrate our framework's effectiveness. Overall, the proposed hierarchical framework appears novel and unique enough to be applied in reinforcement learning. In addition, we designed a specific trajectory planner, Inverse Kinematics solver and trajectory tracking controller for the hexapod robot. This paper makes a valuable contribution to the *Actuators* journal.

The rest of this work is organized as follows. Section 2 introduces the hexapod robot VkHex and the RL-based hierarchical framework. Section 3 describes the design of our framework for the hexapod robot gait generation task. Section 4 presents training details and experiments to verify the effectiveness and feasibility of the proposed framework. Finally, conclusions and avenues for future research are presented in Section 5.

## 2. Robot Prototype and Hierarchical Framework

In this section, we introduce the key points of the physical prototype and gait planning of the hexapod robot VkHex, and propose the corresponding hierarchical architecture based on reinforcement learning.

### 2.1. Hexapod Robot

We designed a hexapod robot with 18 degrees of freedom (DOF) and constructed a simulation environment using the PyBullet [28] physical engine. As shown in Figure 1, VkHex adopts a rectangular structure with six identical legs symmetrically distributed.

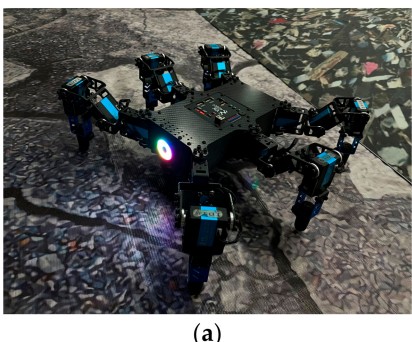
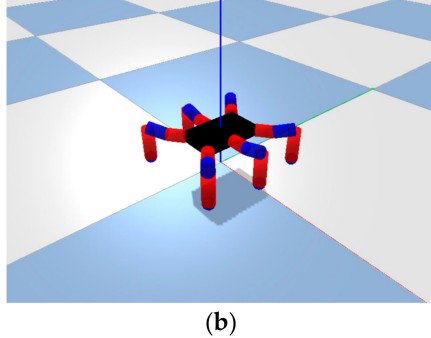

(**a**)                                   (**b**)

**Figure 1.** (**a**) Physical prototype of the hexapod robot VkHex; (**b**) simulated prototype of the VkHex in PyBullet.

The six legs of the hexapod robot are distributed as shown in Figure 2; each leg can be abstracted into a three-link mechanism consisting of hip, knee and ankle joints in series. Table 1 shows details on the VkHex, including dimensions, weight and references.

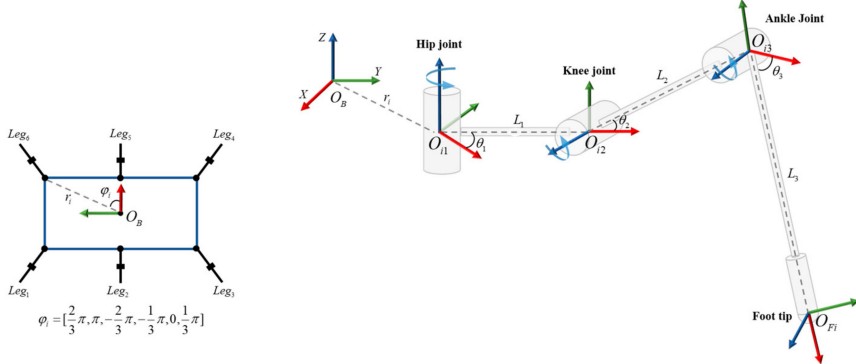

**Figure 2.** The rotation angles of the three active joints relative to the initial position are denoted as $\theta_1$, $\theta_2$ and $\theta_3$. The lengths of the three links are recorded as $L_1$, $L_2$ and $L_3$. The distance from the center of the body to the hip joint of the $Leg_i$ is $r_i$ ($i = 1, \cdots, 6$). A body coordinate frame {$B$} is located in the center of the body. The z-axis of the joint coordinate frame {$OJ_{ij}$} ($j = 1, 2, 3$) coincides with the joint rotation axis. Additionally, the foot tip coordinate frame is denoted as {$OFi$}.

**Table 1.** VkHex details and parameters.

| Parameter | Value | Parameter | Value |
|---|---|---|---|
| Body dimensions | $0.3866 \times 0.2232 \times 0.0821$ m | Robot weight | 10 kg |
| DoF | 18 | Hip link $L_1$ | 0.1179 m |
| Knee link $L_2$ | 0.1302 m | Ankle link $L_3$ | 0.3996 m |
| Servo size | $72 \times 40 \times 56$ mm | Servo weight | 72 g |
| Servo parameters | 27.5 kg/cm 7.4V | Computing device | Nvidia Tx2 |
| 9-axis IMU | MPU9250 | Power | 7.4 V 8000 mAh |

### 2.2. Reinforcement Learning and Hierarchical Framework

Most existing research has applied RL to learn an end-to-end motion controller to control joints directly. However, as the number of robot joints increases, the end-to-end motion controller training becomes difficult and tends to diverge, which is not suitable for a hexapod robot [29]. To address this, we introduced a policy network, gait planner and trajectory planner to design a RL-based modular hierarchical framework. Figure 3 shows the designed modular hierarchical framework based on RL.

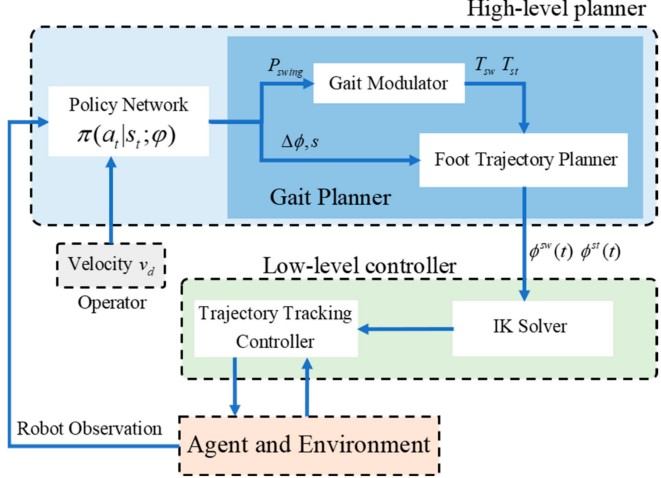

**Figure 3.** Gait generation and motion control architecture based on reinforcement learning and modular hierarchical framework. This architecture decouples the planning and motion control problem of a hexapod robot into two levels, including a motion planner (shown in blue) and joint trajectory controller (shown in green).

We divided the RL-based hierarchical control architecture into a high-level planner and a low-level controller. The high-level planner includes a gait policy network $\pi(a_t|s_t; \varphi)$ and a gait planner. The gait policy network trained by RL outputs the optimal gait parameters to the gait planner, including $P_{swing}$, $\Delta\phi$ and $s$. Then, the gait modulator and foot trajectory planner generate the swing and support phase functions $\phi^{sw}(t)$ and $\phi^{st}(t)$. The Inverse Kinematics (IK) solver and the trajectory tracking controller make up the lower-level controller, and the IK solver converts the foot trajectories into joint angles $\theta_{ij}(i = 1\ldots6; j = 1, 2, 3)$. Finally, the trajectory tracking controller receives joint commands and joint position errors to outputs motor control signals.

*(a)    Gait policy network*

We used a two-layer multi-layer perceptron (MLP) [30] with a tanh activation function to estimate the RL policy function. As shown in Figure 4, the MLP input is a 17D robot observation vector, and the 7D output is directly sent to the gait planner, including the gait phase difference, duty factor and step length.

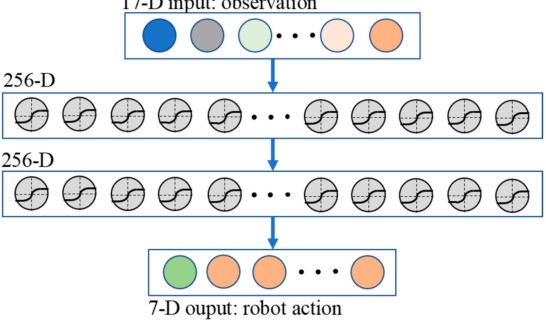

**Figure 4.** Network architecture used for gait policy is a two-layer MLP with tanh activation function. The input is the observation vector of the agent. The output is an extremely simple 7D vector.

*(b)    Gait modular*

When walking on a flat road, hexapod robots alternate between supporting and swing phases to advance the body's motion. In general, the gait duty factor $P_{swing}$ indicates the proportion of the swing phase to the whole gait cycle:

$$P_{swing} = \frac{T_{sw}}{T_{st} + T_{sw}}, \tag{1}$$

where $T_{st}$ is the support time in a gait cycle and $T_{sw}$ is the swing time.

Then, we used a phase function $\phi(t) \in [0, 2)$ and the relative phase differences $\Delta\phi_i$ to parameterize the state of each leg. $\phi(t^{sw}) \in [0, 1)$ indicates that the leg is in the swing phase, and $\phi(t^{st}) \in [1, 2)$ represents the support phase. Where leg-1 is the reference leg and the phase is $\phi_1$, the phases of leg-*i* can be represented by the following:

$$\phi_i = \phi_1 + \Delta\phi_i \quad (i = 2, 3, 4, 5, 6). \tag{2}$$

*(c)    Trajectory planner*

The foot trajectory of a hexapod robot includes the swing and support phase trajectories. In this work, we adopt different phase functions for foot trajectories in three directions:

$$\begin{cases} x = y = s \cdot \phi_x^{sw}(t), \\ z = h \cdot \phi_z^{sw}(t), \end{cases} \tag{3}$$

where $(x, y, z)$ is the position of the foot tip relative to the body coordinate system at time $t$, $s$ is the step length and $h$ is the step length.

When using the cycloid as the swing phase trajectory, the foot tip slides when touching the ground and drags when walking [31]. As shown in Figure 5, we designed a new cycloidal function as the swing phase trajectory:

$$\begin{aligned} \phi_x^{sw}(t) &= \phi_y^{sw}(t) = \frac{t}{T_{sw}} - \frac{1}{2\pi} \sin\left(\frac{2\pi t}{T_{sw}}\right), \\ \phi_z^{sw}(t) &= sgn\left(\frac{T_{sw}}{2} - t\right)\left(2\left(\frac{t}{T_{sw}} - \frac{1}{4\pi}\right) \sin\left(\frac{4\pi t}{T_{sw}}\right) - 1\right) + 1, \end{aligned} \tag{4}$$

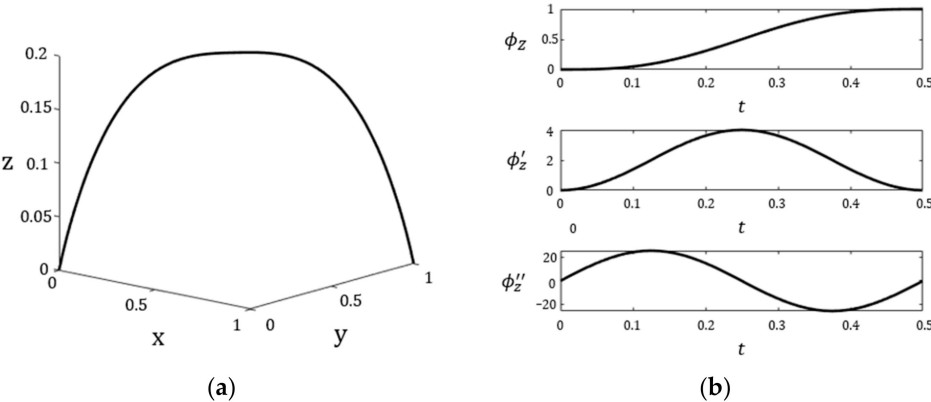

**Figure 5.** Improved cycloidal trajectory (**a**) and cycloidal curve in z-axis (**b**). The modified cycloidal curve eliminates sudden acceleration change in the z-axis direction, as shown in (**b**).

To complete the periodic motion, the robot must satisfy momentum conservation in the vertical direction, but direct force control to ensure the stability of robot motion is extremely difficult [32]. Considering the hexapod robot equilibrium-point hypothesis [33],

we indirectly generated the required support force through position control. Therefore, we used a sinusoidal function as the support phase trajectory:

$$z = -\delta \sin\left(\pi \frac{t}{T_{st}}\right) - \eta \cdot \delta \sin\left(2\pi \frac{t}{T_{st}}\right), \tag{5}$$

where the amplitude $\delta$ is the virtual depth of the foot tip proportional to the support force. Figure 6 shows the support phase trajectories with different parameters.

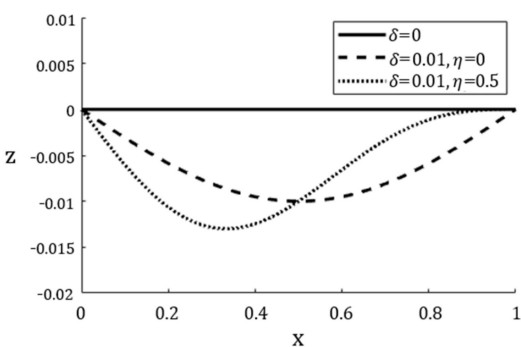

**Figure 6.** The support phase trajectory. In function expression, different amplitudes $\delta$ and $\eta$ can be set to achieve the ideal support phase position control; we adopted $\delta = 0.01$ and $\eta = 0.5$ in this work.

*(d)    Inverse Kinematics solver*

If the position of the foot tip in the hip joint coordinate system $\boldsymbol{P} = \left(^{J_1}x, \, ^{J_1}y, ^{J_1}z\right)$ is known, the IK solver calculates the relationship between the joint rotation angles $\theta_1, \theta_2, \theta_3$ and the foot tip position $\boldsymbol{P}$:

$$\begin{cases} \theta_1 = \tan^{-1}\left(\frac{^{J_1}y}{^{J_1}x}\right), \\ \theta_2 = \cos^{-1}\left(\frac{d_1{}^2 + l_2{}^2 - l_3{}^2}{2d_1 l_2}\right) - \cos^{-1}\left(\frac{d_1{}^2 + d_2{}^2 - {}^{J_1}z^2}{2d_1 d_2}\right) \;, \\ \theta_3 = \cos^{-1}\left(\frac{l_2{}^2 + l_3{}^2 - d_2{}^2}{2l_2 l_3}\right) - \pi, \end{cases} \tag{6}$$

where, $d_1 = \sqrt{^{J_1}x^2 + {}^{J_1}y^2} - l_1, d_2 = \sqrt{d_1{}^2 + {}^{J_1}z^2}$.

*(e)    Trajectory Tracking Controller*

In the hierarchical framework, we adopted a sliding mode controller as the joint trajectory tracking controller and used a nonlinear potential-like function to place the integral:

$$\rho(\lambda \cdot e) = \lambda \frac{e^{\lambda x} - e^{-\lambda x}}{e^{\lambda x} + e^{-\lambda x}} = \lambda \, tanh(\lambda \cdot x), \tag{7}$$

where $\lambda$ is the regulator and $e$ is the joint position error.

Then, a sliding surface with an error integral term was designed to improve the tracking accuracy, reduce the steady-state error, and avoid the difficult convergence problem, expressed as:

$$\boldsymbol{s} = K_P e + K_I \int_0^t \rho(\lambda \cdot e) d\tau + K_D \dot{e} \tag{8}$$

The control law of the nonlinear integral sliding mode control can be expressed as:

$$\dot{\theta} = J_v^{\dagger}(\theta)\left\{K_P^{-1}\left[\hat{\eta}_1 sgn\boldsymbol{s} + \hat{\eta}_2 \boldsymbol{s} + K_I \cdot \rho(\lambda \cdot e) + K_D \ddot{e}\right] + \dot{\phi}(t)\right\}, \tag{9}$$

where $J_v^{\dagger}(\theta) = J_v^T(\theta)\left[J_v(\theta)J_v^T(\theta)\right]^T$ is the pseudo-inverse matrix of the linear velocity Jacobi matrix, $\boldsymbol{s}$ is the sliding surface, $\hat{\eta}_1$ and $\hat{\eta}_2$ are estimates for adaptation laws and $\dot{\phi}(t)$ is the

ideal foot-end velocity trajectory. Figure 7 shows the system block diagram of the nonlinear integral sliding mode controller.

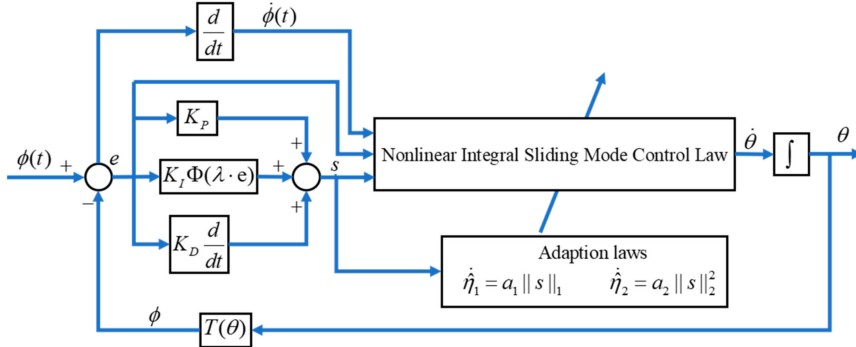

**Figure 7.** The block diagram of the nonlinear integral sliding mode controller.

## 3. Learning Process

This section introduces the learning process for the robot gait generation task, including the environment's Markov decision process, RL model description, and policy training algorithm.

### 3.1. Markov Decision Process

The adaptive gait generate problem of the hexapod robot can be described as a Markov decision process. Here, the MDP is a 5-tuple $\{S, A, r, \Gamma, \gamma\}$, where $S$ is the set of observations, $A$ is the set of actions, $r : S \times A \to \mathbb{R}$ represents the reward given after action and observation, $\Gamma : S \times A \times S \to \mathbb{R}$ is the transition probability function and $\gamma \in (0, 1]$ is the discount factor of the MDP. The robot's goal is to interact with the environment through the optimal policy network that maximizes future rewards.

### 3.2. Action

Unlike common robotic arms and quadruped robots with fewer joints, VkHex has 18 driving joints and 42 alternative support states [34], meaning that the gait has to be searched in an extremely large space. In our framework, we use different parameter groups $\{P_{swing}, \Delta\phi_i, s\}$ to represent different motion gaits, which significantly reduces the dimension of the action space while being intuitive and simple. As a result, in the gait generation task and RL framework, one action includes a 7D vector:

- Gait duty factor $P_{swing}$ (1D);
- Phase difference $\Delta\phi_i$ (5D);
- Step length $s$ (1D).

### 3.3. Observation

The attributes in the observation space consist of only those measurable by VkHex. We categorized the observed attributes into three types: (1) values sensed by the robot, (2) values associated with historical moments, and (3) values related to the designed goals. The observation consists of the following attributes, which add up to a 17D vector:

- Current velocity of the VkHex $v_{base}^{cur}$ (3D);
- Current angular velocity of the VkHex $w_{base}$ (3D);
- Body platform height $H$ (1D);
- Last moment action $a_{t-1}$ (7D);
- Target velocity of the VkHex $v_{base}^{target}$ (3D).

*3.4. Reward Function*

　　Our reward design encourages the robot to generate the most efficient gaits according to different velocity commands while completing the corresponding joint motion control without falling. Therefore, we used one primary reward and four penalties.

(i)　In our study, the gait stability reward $r_{balance}$ is the primary reward. The new generated gaits must be stable enough. Only when the hexapod robot reaches the target position without falling can it receive a positive reward. The functional expression of this reward is as follows:

$$r_{balance} = \begin{cases} \lambda, & \Phi(x_{balance}) \geq 0, \\ -\alpha\lambda, & \Phi(x_{balance}) < 0, \end{cases} \tag{10}$$

where $\lambda > 0$ is the fixed reward value and $\alpha$ denotes discount factor. The function $\Phi(x_{balance})$ judges whether the robot is stable from the termination condition.

(ii)　The learning task is that the hexapod robot can track speed commands and generate new RL-based gaits. The velocity tracking penalty $r_{vel}$ forces the agent to move at the desired velocity:

$$r_{vel} = \exp\left(\|\overline{v}_{base} - v_{base}\|^2\right), \tag{11}$$

where $\overline{v}_{base}$ is the desired velocity and $v_{base}$ is the actual velocity of the robot.

(iii)　The energy consumption penalty $r_{energy}$ is the penalty for gait motion efficiency and energy consumption. We used the cost of transportation (CoT) [35] as the penalty index. The expression is:

$$r_{energy} = \frac{\sum_{i=1}^{18} \tau_i \cdot \omega_i}{mg \cdot |v_{base}|}, \tag{12}$$

where $\tau_i$ is the joint torque, $w_i$ is the joint angular velocity, $m$ is the mass of the robot and $g$ is the gravitational acceleration.

(iv)　The joint tracking penalty $r_{joint}$ is the penalty for the joint tracking error, which aims to improve the joint tracking control accuracy under the premise of stable motion:

$$r_{joint} = \exp\left(-p\|\Delta\theta_i\|^2\right) + \exp\left(-q\|\Delta\dot{\theta}_i\|^2\right), \tag{13}$$

where $\Delta\theta_i$ is the joint position error, $\Delta\dot{\theta}_i$ is the joint velocity error and $p$ and $q$ are the coefficients.

(v)　The roll and pitch penalty $r_{rp}$ penalizes the roll-pitch-yaw angle of the body, which can further improve the stability of RL-based gait:

$$r_{rp} = \exp\left(-\beta\|\omega_{base}\|^2\right), \tag{14}$$

where $\omega_{base}$ is the angular velocity of the robot platform and $\beta$ is the coefficient.

*3.5. Termination Condition*

　　We used an early termination strategy to avoid falling into the local minimum and improve sampling efficiency. If one of the following conditions was met, the agent terminated the training and started again from the initial state:

● 　The robot is involved in a self-collision.
● 　The pitch or roll degree of the base exceeds the allowable range.
● 　The base height is less than the set threshold.
● 　Any link except the foot-tip collides with the ground.

### 3.6. Policy Training

The policy and critic networks are MLPs with two hidden layers each, while the action and observation vectors are the output and input, respectively. We adopted the *Soft Actor-Critic* (SAC) algorithm [36] to maximize the expected reward return:

$$\varphi^* = \underset{\varphi}{\arg\max} E_{\pi_\varphi} \left[ \sum\nolimits_{t=0}^{\infty} \gamma^t \left( r_{balance} + w_1 \cdot r_{vel} + w_2 \cdot r_{energy} + w_3 \cdot r_{joint} + w_4 \cdot r_{rp} \right) \right] \quad (15)$$

where $\gamma$ is the discount factor and $w_i (i = 1, 2, 3, 4)$ is the penalty factor. SAC is an off-policy maximum-entropy DRL algorithm where the actor aims to maximize expected reward and entropy. In the RL framework, SAC provides sample-efficient learning while retaining the benefits of entropy maximization and stability. Algorithm 1 summarizes the essential steps of SAC, where $\lambda_V$, $\lambda_Q$ and $\lambda_\pi$ are the gradients and $\hat{\nabla} J(\cdot)$ are the approximate gradient functions [36]. The specific hyperparameters found empirically in preliminary experiments are shown in Table 2.

---

**Algorithm 1**: Soft Actor-Critic

| | |
|---|---|
| 1 | Initialize policy parameters $\varphi$, replay buffer $D = \{\}$, Soft Q-function parameters $\theta_i$, Soft value function $V$ parameters $\Psi$ and Target critic function $V'$ parameters $\overline{\Psi}$. |
| 2 | **for** iteration = 1, M **do:** |
| 3 |     **for** environment step = 1, N-1 **do:** |
| 4 |         $a_t \sim \pi_\phi(a_t | s_t)$ |
| 5 |         $s_{t+1} \sim \Gamma(s_{t+1} | s_t, a_t)$ |
| 6 |         $D \sim D \cup \{(s_t, a_t, r_t, s_{t+1})\}$ |
| 7 |     **end** |
| 8 |     **for** gradient step = 1, T-1 **do:** |
| 9 |         Update $V$ via minimizing the squared residual error: $\Psi \leftarrow \Psi - \lambda_V \hat{\nabla}_\Psi J_V(\Psi)$ |
| 10 |         Update $Q$ and $Q'$ via minimizing the soft Bellman residual: $\theta_i \leftarrow \theta_i - \lambda_Q \hat{\nabla}_{\theta_i} J_Q(\theta_i)$ for $i \in \{1, 2\}$ |
| 11 |         Update $\pi_\varphi$ via minimizing the expected KL divergence: $\varphi \leftarrow \varphi - \lambda_\pi \hat{\nabla}_\varphi J_\pi(\varphi)$ |
| 12 |         Update $V'$: $\overline{\Psi} \leftarrow \tau \Psi + (1 - \tau) \overline{\Psi}$ |
| 13 |     **end** |
| 14 | **end** |

---

**Table 2.** Algorithm and model hyperparameters.

| Parameter | Value | Parameter | Value |
|---|---|---|---|
| Learning rate | $3 \times 10^{-4}$ | Replay buffer size | $1 \times 10^5$ |
| Discount factor | 0.99 | Entropy regularization | 0.005 |
| Policy network hidden layer nodes | [256,256] | Critic network hidden layer nodes | [400,300] |
| Parameter update frequency | 1 | Gradient update steps | 1 |
| Velocity tracking penalty factor $w_1$ | $-0.1$ | Energy consumption penalty factor $w_2$ | $-0.3$ |
| Joint tracking penalty factor $w_3$ | $-0.1$ | Roll and pitch penalty factor $w_4$ | $-0.35$ |
| Roll and pitch penalty coefficient $\omega$ | 2 | Joint tracking penalty coefficient $m, n$ | 1.5 |

## 4. Experiments and Results

In this section, we designed simulation and comparative experiments to verify the superiority and effectiveness of the RL-based hierarchical framework proposed in this paper.

### 4.1. Implementation Details

Environment bias and modeling uncertainties between the simulation and physical robot affects the porting of the RL model. We adopted the following details in model training to improve training efficiency and model robustness.

(i)    Random model parameters [22]: We used the randomized model parameter strategy, which can improve the policy robustness against modeling errors and noise. Parameters of the robot model were sampled uniformly inside the range provided in Table 3.

**Table 3.** Parameter disturbance range of VkHex model.

| Parameter | Lower Bound | Upper Bound |
| --- | --- | --- |
| Centroid position | −2 cm | 2 cm |
| Link mass | 0.04 kg | 0.06 kg |
| Rotational inertia | 80% | 120% |
| Joint max torque | 80% | 120% |
| Friction coefficient | 0.5 | 1.2 |

(ii)　Introduce sensor noise [30]: All the simulated sensors were noise-free, while the sensors caused data deviation because of the interference and noise in the actual observation. Therefore, we added normal distributed noise to the simulation robot observation and parameters, as shown in Table 4.

**Table 4.** Sensor noise sampling parameters.

| Parameter | Sample Distribution |
| --- | --- |
| Joint position (rad) | $N(0, 0.003)$ |
| Joint velocity (rad/s) | $N(0, 0.03)$ |
| Joint torque (Nm) | $N(0, 0.5)$ |
| Body height (m) | $N(0, 0.01)$ |
| Body velocity (m/s) | $N(0, 0.1)$ |
| Body angular velocity (rad/s) | $N(0, 0.1)$ |

(iii)　External disturbance: Applying random disturbance force has been shown to be effective in achieving sim-to-real transfer and virtual load simulation [17]. During training, the external force was applied to the body from a random direction for every certain number of steps. The disturbance force was generated randomly within $(0, 5N]$ and lasted for 0.5 s.

### 4.2. Training Result

We compared the SAC with some of the algorithms with superior performance, including PPO [37], DDPG [38] and TD3 [39]. Figure 8 shows the learning curves, random sampling and sensor noise cause jitter. Compared with the three algorithms, SAC had the highest learning efficiency and the fastest rise rate in the initial stage. After 150,000 steps, the speed gradually decreased and finally converged. In addition, the convergence of different algorithms also proves the generality of the RL-based hierarchical framework proposed in this paper.

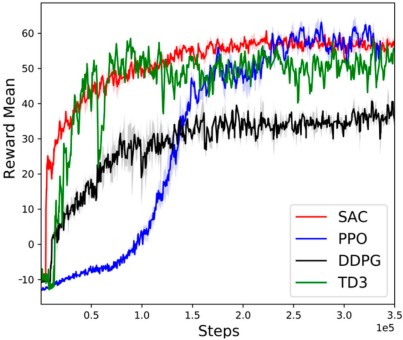 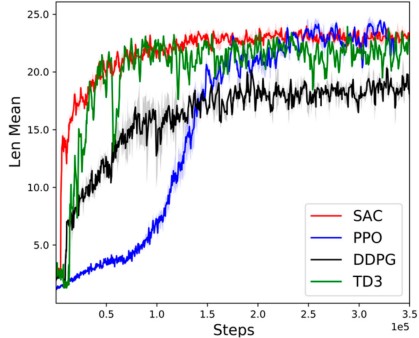

**Figure 8.** The reward curves and learning curves on reinforcement learning hierarchical framework for using SAC algorithm. We stopped training when the parameters were stable. The abscissa represents the number of steps and the ordinate represents the return value (the average epoch length is on the left and the average epoch reward on the right). Our framework learned the gait strategy and converged eventually.

### 4.3. Motion Verification

The initial height of the robot was 0.1 m and the direction was the y-axis of the body coordinate frame. We trained our robot by giving a specific target 1m in front. Once it reached its initial target, the next goal was given again to be 1m ahead repeatedly. We trained the gait policy network in the physics simulation, gradually increasing the expected velocity to 0.6 m/s. Figure 9 shows the motion process of the hexapod robot in seven continuous motion cycles, and Figure 10 shows the corresponding gait phase.

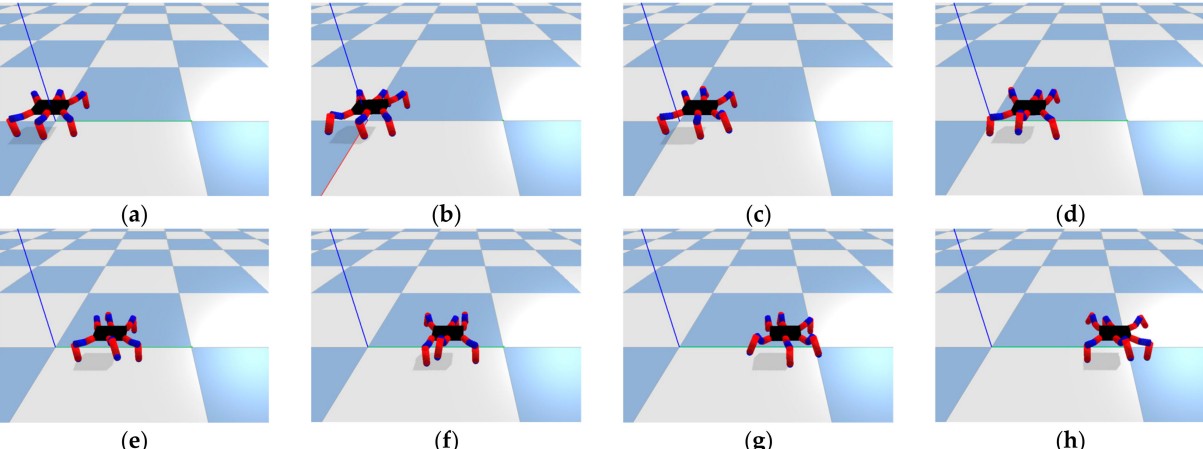

**Figure 9.** Speed-adaptive gaits of the hexapod robot trained in the hierarchical framework for reinforcement learning. From (**a**–**h**), the robot walked in a fixed direction and the motion speed gradually increased to 0.6 m/s. As the speed increased, the hexapod robot gradually transitioned from a wave-like gait to a tripod-like gait, and the robot was stable enough not to fall.

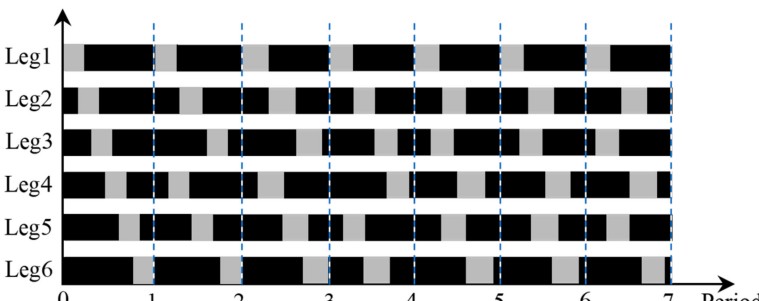

**Figure 10.** Visualization of legs' phase for the RL-based gaits. Black rectangles indicate support phases, gray rectangles indicate swing phases and block lengths indicate duration. The hexapod robot walked with a wave-like gait in the first cycle, a quadruped-like gait in the second to fourth cycles, and finally walked in a tripod-like gait. Within each gait cycle, RL-based gait had a different duty cycle and phase difference.

The results show that the gait strategy network can generate new gaits according to different velocity commands and motion velocity. The RL-based gait is similar to the tripod, quadruped and wave gaits, but the RL-based gait phase difference and duty factor differ. In addition, all these gaits were both new and stable. Since the reward function of the RL-based framework is dominated by stability, the swing phase took less time in the RL-based gait circle to achieve stability.

### 4.4. Motion Efficiency Comparison

Referring to Equation (10), we compared the transportation cost of the hexapod robot under the RL-based gaits and three rhythmic gaits. The robot motion gait period was set to 10, and the motion velocity was 0.1 to 1.1 m/s. We added up the transportation cost of all periods to be the total CoT and repeated the same experiment three times. Figure 11 shows the mean transportation cost and standard deviation (S.D.) of the hexapod robot in the four gaits.

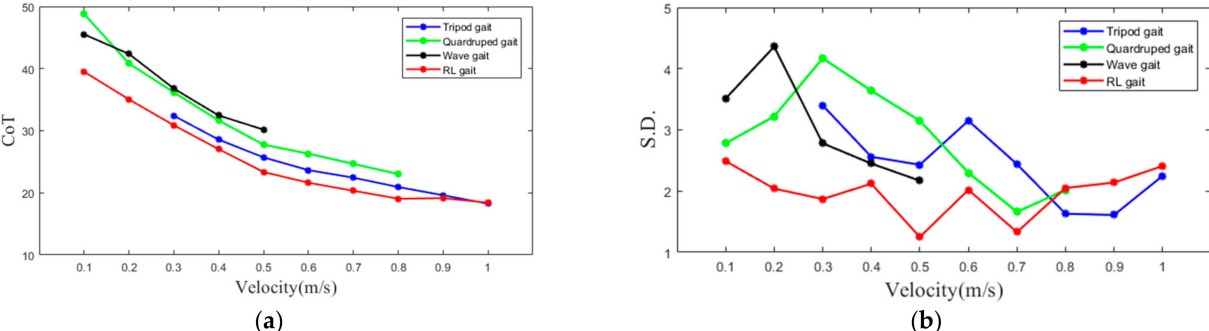

(**a**)                                                  (**b**)

**Figure 11.** The mean CoT (**a**) and standard deviation (**b**) of the hexapod robot in four different gaits. Considering structural features and motion stabilization, a hexapod animal would not use a wave gait for fast walking and a tripod gait for slow walking.

Comparing three rhythmic gaits, the RL-based gait had the lowest CoT and the highest motion efficiency. Additionally, the RL-based generated in our framework was speed-adapted. In the reward, the energy consumption penalty factor was second only to the gait stability reward, so the RL-generated gaits considered both gait stability and energy consumption. The experimental results show that the RL-based gait was superior to the three rhythmic gaits in motion efficiency.

*4.5. Sim-to-Real*

The previous simulations illustrate that gait policy network can be trained by our framework and is optimized enough to perform adaptive and stable gaits. In this experiment, we deployed the same policy on the physical VkHex robot and the policy ran 100 Hz on VkHex. As the movement speed increased from 0m/s to 0.5 m/s, the RL-based gaits of the hexapod robot behaved as shown in Figure 12. Compared with fixed rhythmic gait, the RL-based gaits performed more flexibly. As speed increased, VkHex could adaptively generate rhythmic-like gaits. Furthermore, all gaits were stable during locomotion, which, when performed in the real world, were similar to the simulated RL-based gaits. Since the environmental disturbances and modeling uncertainties between simulation and reality are different, such as servo force, friction coefficient, sensor noise and inaccurate motor models, the variety and stability of the gait in reality was slightly poorer than in simulation. The results of the simulated and real experiments further illustrate the effectiveness of the proposed framework.

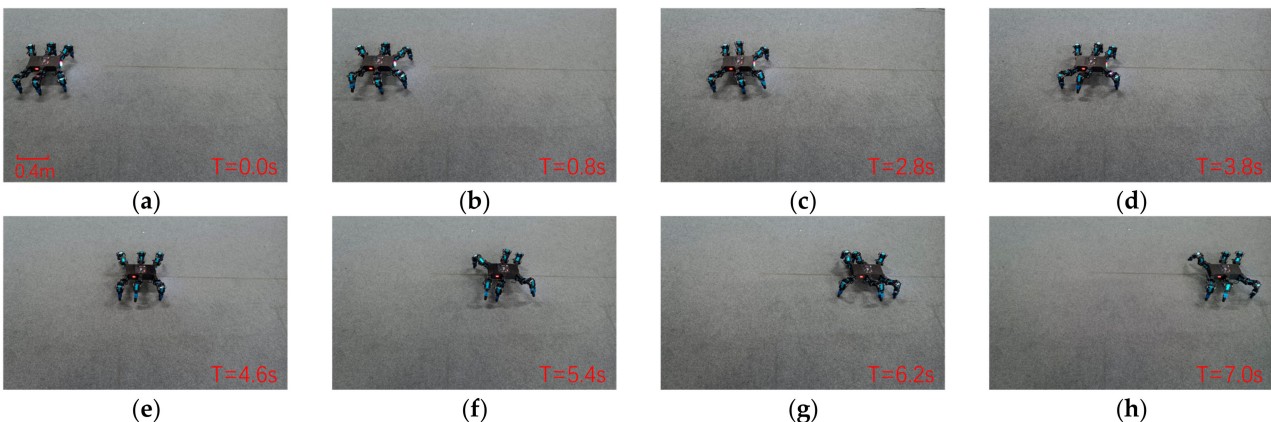

**Figure 12.** RL-based hierarchical framework tested on the VkHex prototype on flat terrain. Similarly, from (**a**–**h**), the robot walked in a fixed direction while the motion speed gradually increased to 0.5 m/s. The hexapod robot walked primarily in a triangular-like gait and slightly to the left.

## 5. Conclusions

In this paper, a RL-based hierarchical framework is proposed to simplify the hexapod action and learn a RL-based gait generation task, including a policy network, gait planner, IK solver and trajectory tracking controller. Furthermore, to train an adaptive gait generation policy network, we designed a whole RL framework for the hexapod robot and tested our framework using SAC, PPO, DDPG and TD3 in the physics simulation. Our framework enabled the DRL algorithms to converge and allowed the hexapod to generate adaptive new gaits on flat terrain. Finally, we ran the same policy in the real robot without any modification. Through comparison and experiments, it was verified that this framework is effective for RL tasks. This paper realizes the speed-adaptive gait generation and planning of the hexapod robot. We only tested using flat terrain, so further research could include a study of the similarities and differences in reward design under the RL framework for challenging terrains and tasks.

**Author Contributions:** All authors contributed to this work. Conceptualization, Z.Q.; methodology, Z.Q.; software, Z.Q.; validation, Z.Q. and W.W.; writing—original draft, Z.Q.; writing—review and editing, Z.Q. and X.L. All authors have read and agreed to the published version of the manuscript.

**Funding:** This work was supported by the National Natural Science Foundation of China [Grant Nos. 61573148, 61603358] and the Science and Technology Planning Project of Guangdong Province, China [Grant Nos. 2015B010919007, 2019A050520001].

**Institutional Review Board Statement:** Not applicable.

**Informed Consent Statement:** Not applicable.

**Data Availability Statement:** Not applicable.

**Conflicts of Interest:** The authors declare no conflict of interest. The funders had no role in the design of the study; in the collection, analyses, or interpretation of data; in the writing of the manuscript, or in the decision to publish the results.

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
