# Peer review of "Adaptive Gait Generation for Hexapod Robots Based on Reinforcement Learning and Hierarchical Framework"

_actuators, doi:10.3390/act12020075_

Round 1
Reviewer 1 Report
The article “Adaptive Gait Generation for Hexapod Robots Based on Reinforcement Learning and Hierarchical Framework” proposes an interesting approach for a hexapod control system that is learning different gaits in order to show more adaptive behavior. Overall, I see this as a valuable contribution for the actuators journal and think that the presented concept is a nice contribution and the proposed hierarchy appears novel or unique enough for how it is applied in reinforcement learning. But there is still quite some work left to do how the results are presented and related work is provided. Therefore, I would recommend major revision for the manuscript and will provide more details in the following.
First, the authors should better frame their approach in context of related work. From my point of view, there are two things where this paper is lacking: On the one hand, for learning-based approaches the authors should provide more related work. They nicely point out the important work by Hwangbo, Hunter and colleagues. But there has already been quite some follow-up work from the same group. Or the recent Nature Machine Intelligence article by Manoonpong - Thor, M., & Manoonpong, P. (2022). Versatile modular neural locomotion control with fast learning. Nature Machine Intelligence, 4(2), 169-179 (see as well an overview article on biological inspired legged approaches from the same author Manoonpong, Poramate, Luca Patanè, Xiaofeng Xiong, Ilya Brodoline, Julien Dupeyroux, Stéphane Viollet, Paolo Arena, and Julien R. Serres. “Insect-Inspired Robots: Bridging Biological and Artificial Systems.” Sensors 21, no. 22 (January 2021): 7609. https://doi.org/10.3390/s21227609.).
On the other hand, while the authors motivate their approach to some degree from biology, I found this part in particular lacking and as the authors are in particular trying to differentiate their work from CPG-based control and point towards different gaits and transitions between these, I think the authors might want to add work following the Walknet based approach as one example for an emergent gait controller (e.g. Schilling et al., “Walknet, a Bio-Inspired Controller for Hexapod Walking.”; or with a relation to learning the more recent papers on DRL as Schilling et al. (2021), “Decentralized Control and Local Information for Robust and Adaptive Decentralized Deep Reinforcement Learning.” or applied on a six-legged robot Schilling, Konen, Ohl & Korthals (2020). “Decentralized deep reinforcement learning for a distributed and adaptive locomotion controller of a hexapod robot”; or work as Owaki, D., & Ishiguro, A. (2017). A quadruped robot exhibiting spontaneous gait transitions from walking to trotting to galloping. In Scientific reports. http://dx.doi.org/10.1038/s41598- 017- 00348- 9.).
Furthermore, there is in addition some work on hierarchical DRL that should be added. On the one hand, quite generally, e.g. the excellent overview Merel, Botvinick, and Wayne, “Hierarchical Motor Control in Mammals and Machines.”; the recent overview by Eppe et al., “Intelligent Problem-Solving as Integrated Hierarchical Reinforcement Learning.”; or applied to robots more concretely Azayev and Zimmerman, “Blind Hexapod Locomotion in Complex Terrain with Gait Adaptation Using Deep Reinforcement Learning and Classification.” or wrt. emergent gaits Schilling and Melnik, “An Approach to Hierarchical Deep Reinforcement Learning for a Decentralized Walking Control Architecture.”).
I found this part of the introduction should be much more fleshed out - and maybe become its own section.
Secondly, as one further major criticism: It is not completely clear what the authors aim for and how much this relates to such more biological control approaches. The authors talk about gaits, but their understanding remains a little bit fuzzy. They (and I would agree with this idea) mostly relate a gait to phase relations between legs. But then it appears that they reproduced each gait with quite different velocities. That is at least surprising and should be motivated. In animals it appears that different phase relations are a consequence of different velocities and it is not clear why a six-legged animal, as an example, would use a tripod gait with a very slow velocity (here, a gait would be directly related to a duty factor and for a slow tripod gait this would result the animal making very slow swing movements which is not advantageous at all). Furthermore, the authors mention gait transitions, but these are not shown in the results. Overall, I found this quite unclear: as the authors try to differentiate their work from CPG-like approaches, they should state what they mean by gait, e.g. relating to work as for example DeAngelis et al (2019) “The Manifold Structure of Limb Coordination in Walking Drosophila.” ELife 8; or Bidaye et al., “Six-legged walking in insects: How CPGs, peripheral feedback, and descending signals generate coordinated and adaptive motor rhythms" which demonstrate that in slow walking in insects, gaits are more an observation as an underlying control structure.
Third, the result section requires some work and should be better organized plus extended. I have quite a lot detailed question and comments (see below), but I would first propose to restructure this completely and show much more results. Currently, these appear quite selected and while they show some data or a figure, this is immediately quite broadly interpreted. In general, I would ask to reconsider the results section: the results appear to be there - please show these (in figures, tables). What I would like to see:
- Rewards / returns - and relation to velocity = show good tracking of velocity (this was the goal)
- show that these are proper walking behaviors: gait pattern, but somehow measure stability as well (plus ideally movies)
- efficiency comparison
- demonstration, application on robot
Parts of these are there, but this should be better organized (see as well comments below).
Overall, the language is OK, but could be improved.
Detailed comments for specific sections:
ABSTRACT
While I like that the authors quickly make their point, this could be better framed. Start with the overall goal and challenges, but rephrase this section. The overview of the article should be moved to the end of the introduction, not be part of the abstract.
INTRODUCTION
Line 32 - “requiring human intervention to achieve gait transition.” I don’t understand this sentence - or disagree for what I understand.
l. 37 and following - I found these sentences and explanations to brief. For CPG - you should provide more details and references for some, quite strong statements.
l. 68 - MDP (one P too much)
2 ROBOT
2.A Please provide details on the robot: either reference or some details on servos, strengths, dimensions, weight (maybe a table in Appendix).
l. 86 - I think this should be the other way round: it is proportion of stance P_stance as given in the formula.
l. 101 - Why are the lower levels not learned? Plus: Show plot of the function - this should be a simple trajectory in each dimension (over time), easily visualized through a plot.
2.B I agree with the statement on difficulties of scaling, but you should provide a reference for this.
Section from line 118 onwards: This is a little fuzzy: they state similar things in introduction. I would appreciate to be more precise. In particular to understand how this differs. In a way, many current RL approaches on legged robots use a form of hierarchy, i.e. RL finds positional targets and lower level computes joint movements or torques using MPC. So, specify explicitly input and outputs. In particular: You emphasize the gait modulator, but this is barely described here.
Line 127 - I don’t understand this first sentence.
3 LEARNING
Line 135 remove the “would”
Line 146 - Reference is unclear: Which quadruped or robotic arm?
C - You only use sensors that can be used on the robot. But why not joint state / position / torque?
D - Reward: I found the ordering confusing. I would have expected as a primary reward the velocity tracking.
Please, also refer to other publications and compare rewards.
Line 192: Why is i=24 and not 18? This should be the number of joints or not?
4 EXPERIMENTS
For domain randomization - add reference, e.g. Tan et al (2018), Sim-to-Real, Mozifian et al (2019), Learning Domain Randomization Distributions for Training Robust Locomotion,
Line 218 - please rephrase, e.g. Parameters of the robot model are sampled uniformly inside the range provided in table 2.
From line 220: sensor noise is also quite commonly used, please provide a reference.
In addition, noise and standard deviation are hard to access given on its own. You should provide std. dev. during normal operation (without added noise) in the input channel as a comparison.
In particular, these values seems very low. Probably a lot of real sensors would not even have such a resolution.
Fig. 4 and text: The text says that this shows that it “ is suitable for gait generation”. But you don’t show gaits here, but mean reward (probably return?). First, this has to be set into relation e.g. velocity of robot. Second, in order to discuss gaits you should more data: gait patterns (as one is given), video and something like body clearance or similar which demonstrates that a form of locomotion is established.
Fig. 6: This is nice and helpful. This is what I would like to see as well in the section on simulation (plus video), but also provide names of legs (1 to 6 is ambiguous).
Line 239: Again, I found this statement too strong. There is one Figure showing one gait and a couple of stills, but it is claimed that gaits are optimal and for different velocities. I don’t agree with this: Optimality is probably not possible to show - only better wrt. Efficiency could be done but was not shown here. And you should show different velocities in a table.
Table 4:
I don’t really understand what RL-based gait means. From my understanding this is tracking a given velocity. Tripod, tetrapod etc. are characterizations based on gait patterns - these are based on observing the behavior. So, the RL-based gait should produce (depending on velocity) such patterns - and Fig. 6 shows one such example.
But the authors show gaits for different velocities - this is strange and not what we would expect in animals. In slow walking, animals use wave gaits. But we wouldn’t observe this in faster walking. And, the other way round, you wouldn’t use a tripod gait for slow walking - a tripod is characterized by its duty factor. And when walking slowly it would be detrimental (as, e.g., the slow velocity is making you loose stabilization from momentum) and there is no need to keep legs in swing and lifted from the ground for such very long times.
Line 265 and 269: I found these statements too strong. They are not backed up by the authors table and they don’t show other controllers.
Author Response
Manuscript No. 2113764
Adaptive Gait Generation for Hexapod Robots Based on Reinforcement Learning and Hierarchical Framework
Responses to Reviewer#1
Dear Reviewer
Thank you for giving us an opportunity to revise our manuscript! Now we are submitting the revised manuscript entitled “Adaptive Gait Generation for Hexapod Robots Based on Reinforcement Learning and Hierarchical Framework” for consideration for publication in Sensors. Following your suggestions, we have revised and re-organized the paper carefully, all revises in manuscript we have highlighted it in blue. For details, please see the following answers.
Comment 1: the authors should better frame their approach in context of related work. From my point of view, there are two things where this paper is lacking: On the one hand, for learning-based approaches the authors should provide more related work. On the other hand, while the authors motivate their approach to some degree from biology, I found this part in particular lacking and as the authors are in particular trying to differentiate their work from CPG-based control and point towards different gaits and transitions between these. There is in addition some work on hierarchical DRL that should be added, this part of the introduction should be much more fleshed out - and maybe become its own section.
Response: Thank you for your helpful analysis and suggestions. We have made corrections and rephrased this section to address your suggestions. Firstly, we have added related work for reinforcement learning approaches in robot control. Secondly, to draw out our approach, we have introduced the CPG and Walknet approach and work to achieve a rhythmic behavior. Lastly, we used a new section to introduce some work on hierarchical DRL for robot locomotion. The modified comparison is shown as below:
|
Locomotion in legged animals is characterized as a rhythmic behavior [12]. To achieve a rhythmic gait, some existing studies used Central Pattern Generators (CPG), as an open-loop oscillator [13], which rely on centrally generated rhythms to drives the overall behavior. Support for such approaches has been found in particular on fast walking in insects. Nice examples are the rapid locomotion in cockroaches [14] or running and swimming behavior of the salamander [15]. Importantly, with the high number of joints this becomes a challenging problem and open-loop oscillators solutions are usually not applicable [16]. This brought these lines of research together and has led to biologically-inspired control approaches. As one example, the Walknet approach for hexapod robots realizes a decentralized and modular structure that reflects insights from walking in stick insects [17]. While this approach can deal with a variety of disturbances during locomotion, it is still limited dealing with novel and challenging walking situations [16]. In recent years, data-driven-based methods have attracted significant attention because of their capability of making more accurate and robust control policies, which provides new feasible schemes for hexapod robot motion control [18]. The Walknet approach for hexapod robots realizes a decentralized and modular structure. Walknet can deal with a variety of disturbances during locomotion, it is still limited dealing with novel and challenging walking situations. As a type of data-driven-based method, reinforcement learning (RL) can learn feasible planning and control strategies from data and trial. RL for legged robot walking task has been extensively studied and applied. For instance, Peng et al. proposed a deep learning optimization strategy to train a biped robot in simulated animations, enabling the robot to pass through random obstacles freely [19]. In addition, Tan used proximal policy optimization to train a quadruped robot motion strategy, and realized the transformation from simulation to physical robot [20]. Similarly, Tsounis constructed the Markov decision process (MDP) and planned trajectories in the high-dimensional, continuous, state-action space, realizing the stable movement of the robot in different environments [21]. In relevant research, Deep Reinforcement Learning (DRL) has proven itself capable of automatically acquiring control policies to accomplish a large variety of challenging locomotion tasks. Fu et al. designed a novel DRL method to implement multi-contact motion planning for hexapod robots moving on uneven plum-blossom piles [22]. Specially, Thor and Manoonpong proposed a simple yet versatile modular neural control framework with fast learning, in which behavior-specific control modules can be added incrementally to obtain increasingly complex emergent locomotion behaviors [23]. Impressive performance of DRL for special tasks in footed robots. Miki employed a model-free reinforcement learning approach to train a deep policy which enables the ANYmal robot to balance a light-weight ball robustly using its limbs without any contact measurement sensor [24]. As a series-parallel composite omnidirectional mobile vehicle, the hexapod robot is a complex coupling agent. As a result, using DRL to train an end-to-end controller leads to non-convergence and gait disorder during training [25]. This has led to a growing interest in hierarchical control frameworks and how these frameworks could be exploited to improve behaviors in DRL. For example, Merel et al. proposed several such bio-inspired principles of hierarchical control and advocated their implementation also into robot architectures [26]. As one example, they emphasize the hierarchical framework with higher-level planner modulate the lower-level controller. In addition, Eppe et al. provide the cognitive foundations of hierarchical problem-solving and proposed steps to integrate biologically inspired hierarchical mechanisms to enable advanced problem-solving skills in artificial agents [27]. Both above two works provided detailed and excellent overview for hierarchical reinforcement learning, which was significantly to our paper. Similarly, Azayev and Zimmerman trained policies independently in individual scenarios using DRL, and presented a scalable two-level hierarchical for hexapod locomotion through complex terrain without the use of exteroceptive sensors [1]. |
Comment 2: It is not completely clear what the authors aim for and how much this relates to such more biological control approaches. The authors talk about gaits, but their understanding remains a little bit fuzzy. They (and I would agree with this idea) mostly relate a gait to phase relations between legs. But then it appears that they reproduced each gait with quite different velocities. That is at least surprising and should be motivated. In animals it appears that different phase relations are a consequence of different velocities and it is not clear why a six-legged animal, as an example, would use a tripod gait with a very slow velocity (here, a gait would be directly related to a duty factor and for a slow tripod gait this would result the animal making very slow swing movements which is not advantageous at all). Furthermore, the authors mention gait transitions, but these are not shown in the results. Overall, I found this quite unclear: as the authors try to differentiate their work from CPG-like approaches, they should state what they mean by gait.
Response: Thanks very much for your kind work and valuable suggestions on our paper. In nature, hexapod animals use wave gaits in slow walking and tripod gaits in faster walking, they can choose the right gait according to walking speed. In this paper, we aim to achieve such performance on a hexapod robot by using reinforcement learning methods and biological hierarchical frameworks. Follow your suggestions, we have rephrased the corresponding content. Besides, we apologize for our misrepresentation and are not trying to differentiate our work from CPG. CPG and Walknet are existing methods to achieve rhythmic behavior, we use they to introduce the reinforcement learning method and my framework
Comment 3: The result section requires some work and should be better organized plus extended. I have quite a lot detailed question and comments (see below), but I would first propose to restructure this completely and show much more results. Currently, these appear quite selected and while they show some data or a figure, this is immediately quite broadly interpreted. In general, I would ask to reconsider the results section: the results appear to be there - please show these (in figures, tables). What I would like to see: 1)Rewards / returns - and relation to velocity = show good tracking of velocity (this was the goal) ;2)show that these are proper walking behaviors: gait pattern, but somehow measure stability as well (plus ideally movies) ;3)efficiency comparison ;4)demonstration, application on robot
Response: Thanks very much for your kind work and consideration on our result. We have restructured and revised this section in revised manuscript.
Comment 4: While I like that the authors quickly make their point, this could be better framed. Start with the overall goal and challenges, but rephrase this section. The overview of the article should be moved to the end of the introduction, not be part of the abstract.
Response: Thanks for your valuable and insightful suggestions. Following the above guide, we have removed the overview of the article and rephrased this section. The improved abstract is displayed below.
|
In general, the hexapod robot walks with a fixed rhythmic gait, which reduces its adaptability and limits motion efficiency. Moreover, the hexapod robot has a high-dimensional action space, it has a great challenge to use reinforcement learning to directly train the robot's joint angles. As a result, a hierarchical and modular framework and learning details are proposed in this paper, using only seven-dimensional vectors to denote the agent actions. In addition, we conduct experiments and deploy the proposed framework to a real hexapod robot. The experimental results show that superior reinforcement learning algorithms can converge in our framework, such as SAC, PPO, DDPG and TD3. Specifically, the gait policy trained in our framework can generate new adaptive hexapod gait on flat terrain, which is stable enough and has lower transportation cost than rhythmic gaits. |
Comment 5: Line 32 - “requiring human intervention to achieve gait transition.” I don’t understand this sentence - or disagree for what I understand.
Response: Thanks for your suggestions. I’m sorry for my unreasonable presentation, what we want to express is that the hexapod robot walks by gaits and gait transition. We have carefully checked the expressions of the whole paper and revised it.
|
From the view of robotic motion, a hexapod robot relies on the alternate support and swing of each limb to advance the body’s motion [11], the movement of the hexapod robot is constrained by gaits. |
Comment 6: In the line 37 and following - I found these sentences and explanations to brief. For CPG - you should provide more details and references for some, quite strong statements.
Response: We sincerely appreciate your thoughtful suggestions, we have revised this part.
Comment 8: In the line 68, MDP (one P too much)
Response: Thanks for your suggestions, we have removed it and corrected the punctuation.
Comment 9: 2.A Please provide details on the robot: either reference or some details on servos, strengths, dimensions, weight (maybe a table in Appendix).
Response: Thanks for your guide. We have provided the relevant details of the hexapod and servos in Table 1. The added comparisons are shown as below:
|
The six legs of the hexapod robot are distributed as shown in the Figure 2, each leg can be abstracted into a three-link mechanism consisting of hip, knee, and ankle joints in series. Table 1 shows details on the VkHex, including dimensions, weight and references.
Table 1 VkHex details and parameters
|
Comment 10: In the line 86 - I think this should be the other way round : it is proportion of stance P_stance as given in the formula.
Response: Thanks for pointing out my mistake, I’m sorry for my carelessness. Meanwhile, we have carefully checked the formulations of the whole paper and revised it.
|
|
Comment 11: In the line 101 - Why are the lower levels not learned? Plus: Show plot of the function - this should be a simple trajectory in each dimension (over time), easily visualized through a plot.
Response: Thanks very much for your suggestions. The lower level includes the IK solver and the trajectory controller, which we have added and introduced in Section 2.B.d) and 2.B.e). Meanwhile, we show plot of the function trajectory over time as follows.
|
|||||||||||||||||
Comment 12: 2.B I agree with the statement on difficulties of scaling, but you should provide a reference for this.
Response: Thank you very much for your comments, we have revised it and provided corresponding references.
Comment 13: Section from line 118 onwards: This is a little fuzzy: they state similar things in introduction. I would appreciate to be more precise. In particular to understand how this differs. In a way, many current RL approaches on legged robots use a form of hierarchy, i.e. RL finds positional targets and lower level computes joint movements or torques using MPC. So, specify explicitly input and outputs. In particular: You emphasize the gait modulator, but this is barely described here.
Response: Thanks very much for your valuable and insightful suggestions. I'm sorry for my repeated introduction. We have specified each module in the hierarchical control framework, especially added explicitly inputs and outputs. In additional, we have described the whole framework in Section 2.B.a) ~ 2.B.e), including gait modulator, trajectory planner, IK solver and tracking controller.
Comment 14: Line 127 - I don’t understand this first sentence.
Response: Thanks very much for your question. For some simple reinforcement learning strategies, a table can be used to list the correspondence between all observations and actions, which is not applicable to hexapod robot gait generation. Because the gait strategy (correspondence between observation and action) is not deterministic and cannot be represented in a finite table, we use a MLP to estimate the RL policy function. To avoid misunderstanding readers, we have removed this part, we directly introduce that a MLP is used to estimate the RL policy function.
Comment 15: Line 135 - remove the “would”
Response: Thanks for your suggestions, we have removed it and thoroughly checked the punctuation.
Comment 16: Line 146 - Reference is unclear: Which quadruped or robotic arm?
Response: Thank you for your helpful advice. In general, the robotic arm has 6 joints and the quadruped robot has 12, while the hexapod robot has more driven joints, we want to make a simple comparison here to illustrate hexapod robot has 42 alternative support states, and we use only a 7D vector to represent the action. I have modified the representation of this part, thanks for your suggestions.
Comment 17: C - You only use sensors that can be used on the robot. But why not joint state / position / torque?
Response: Thank you for asking the question and giving us the opportunity to explain it. Some papers add joint state, position and torque to the observation vector, which apparently improves the locomotion of the hexapod. However, as the dimension of the observation space increases, the training time and difficulty of the policy network will increase. In this work, our goal is to realize the adaptive gait generation of the hexapod robot while simplifying the observation vector and learning process as much as possible.
Comment 18: Reward: I found the ordering confusing. I would have expected as a primary reward the velocity tracking. Please, also refer to other publications and compare rewards.
Response: Thanks for your question. In this work, we hope the RL-based gait is stable enough while the robot walks without fall or roll over. This is the premise, so we use the stability reward as a primary reward, followed by the speed tracking reward and energy consumption. In addition, the model hyperparameters in Table 2 has been tested and meets the task requirements. Referring to other publications and rewards, we have optimized the structure and description of Reward.
Comment 19: In the line 192: Why is i=24 and not 18? This should be the number of joints or not?
Response: Thank you very much for pointing out the error, i=18 is correct and corresponds to the number of joints, we have revised it.
Comment 20: For domain randomization - add reference, e.g. Tan et al (2018), Sim-to-Real, Mozifian et al (2019), Learning Domain Randomization Distributions for Training Robust Locomotion. In the line 218, please rephrase, e.g. Parameters of the robot model are sampled uniformly inside the range provided in table 2.
Response: Thanks very much for your valuable and insightful recommendations. We have revised and rephrased it.
Comment 21: From line 220: sensor noise is also quite commonly used, please provide a reference. In addition, noise and standard deviation are hard to access given on its own. You should provide std. dev. during normal operation (without added noise) in the input channel as a comparison. In particular, these values seems very low. Probably a lot of real sensors would not even have such a resolution.
Response: Thanks for your detailed comments, and we have provided a relevant reference. We add normal distributed noise in simulation, and noise and standard deviation are found empirically in preliminary experiments.
Comment 22: Fig. 4 and text: The text says that this shows that it “is suitable for gait generation”. But you don’t show gaits here, but mean reward (probably return?). First, this has to be set into relation e.g. velocity of robot. Second, in order to discuss gaits you should more data: gait patterns (as one is given), video and something like body clearance or similar which demonstrates that a form of locomotion is established.
Response: Special thanks for your helpful suggestions, we are sorry for our error expression. In Fig. 4 and text, we wanted to prove the effectiveness of our framework and shown the RL learning process. We have rephrased this section and trained our framework using superior DRL algorithms.
|
We compare the SAC with some of the algorithms with superior performance, including PPO [37], DDPG [38] and TD3 [39]. Figure 8 shows the learning curves, random sampling and sensor noise cause jitter. Compare with three algorithms, SAC has the highest learning efficiency and the fastest rise rate in the initial stage. After 150,000 steps, the speed gradually decreases and finally converges. In addition, the convergence of different algorithms also proves the generality of the RL-based hierarchical framework proposed in this paper.
|
||||
Comment 23: Fig. 6: This is nice and helpful. This is what I would like to see as well in the section on simulation (plus video), but also provide names of legs (1 to 6 is ambiguous).
Response: Thanks for your detailed suggestions. We have provided the names of legs and coordinate frames in Figure 2.
Comment 24: Line 239: Again, I found this statement too strong. There is one Figure showing one gait and a couple of stills, but it is claimed that gaits are optimal and for different velocities. I don’t agree with this: Optimality is probably not possible to show - only better wrt. Efficiency could be done but was not shown here. And you should show different velocities in a table.
Response: Thanks for your suggestions. I’m sorry for my unreasonable presentation. Figure 10 shows the RL-based gait strategy can generate new gaits, we have revised it.
Comment 25: I don’t really understand what RL-based gait means. From my understanding this is tracking a given velocity. Tripod, tetrapod etc. are characterizations based on gait patterns - these are based on observing the behavior. So, the RL-based gait should produce (depending on velocity) such patterns - and Fig. 6 shows one such example. But the authors show gaits for different velocities - this is strange and not what we would expect in animals. In slow walking, animals use wave gaits. But we wouldn’t observe this in faster walking. And, the other way round, you wouldn’t use a tripod gait for slow walking - a tripod is characterized by its duty factor. And when walking slowly it would be detrimental (as, e.g., the slow velocity is making you lose stabilization from momentum) and there is no need to keep legs in swing and lifted from the ground for such very long times.
Response: Thank you for asking the question and pointing out our mistakes. In this paper, we used reinforcement learning to train hexapod agent’s gait generation policy, so RL-based gait is the new gait generated in our reinforcement learning hierarchical framework with different phase difference and duty factor. In section 4.D, we compared the transportation cost of the hexapod robot under the RL-based gaits and three rhythmic gaits. We are sorry for not consider the characteristics of animal behavior. In general, hexapod use wave gaits in slow walking and use tripod gaits in faster walking, we have revised Table 4 according to your suggestion
Once again, sincere thanks for your time and effort in further processing our revised paper. Looking forward to your guidance again.

Reviewer 2 Report
The paper reports on reinforcement learning (RL) method deployment on the virtual and real hexapod. The authors propose a low-dimensional gait representation which is then tuned by actor-critic policy training in a simulated environment. The policy is then slightly tested in the real environment.
Summary
I like the idea of the parametric representation of the gait combined with actor-critic policy training, which is possible to empirically evaluate with sim-to-real deployment on the hexapod robot. However, the state-of-the-art is not well represented and the claimed method properties are not experimentally corroborated.
Framing
Throughout the paper, the proposed RL-based method is put to contrast against CPG-based methods, which are, in my opinion, misrepresented. The dichotomy between CPG and RL-based methods (lines 55-58) is false as they can be combined into one method which adapts to the environment [1]. Moreover, the CPG is presented as an open-loop system (lines 37-39), which is not true as CPG can synchronize with the environment and select the gaits as a response to the environment [2]. From my understanding of the paper, the proposed method should be contrasted with the idea of open-loop controllers.
However, rather than discussing the difference between RL-based and open-loop controllers, it would be more interesting if the authors discussed the differences between the proposed and other RL-based controllers.
Experimental evaluation
Claims about the proposed algorithm are not corroborated statistically or not tested at all. In the abstract, four hypotheses are stated: proposed controller (i) is stable enough, (ii) has lower CoT than open-loop gaits, (iii) is effective, and (iv) is feasible. However, the experiments test only the second hypothesis (the proposed controller has lower CoT than open-loop gaits), and the reported results lack statistics thus I don't know whether the results are significant. In the other hypotheses (i, iii), the metrics of stability and effectivity are missing and it is not clear whether the authors compare it to the general state-of-the-art or just to the three open-loop gaits. (Minor note: The feasibility property in the fourth hypothesis might not be important as it is implied by other hypotheses.)
The hypotheses should have clearly defined metrics and the results should show statistics of the metrics. The results should be ideally collected on the real hexapod robot.
Other remarks
- Section 2.A. should include the description of the sensory hardware that composes the observation vector described in 2.C.
- It should be mentioned that the hyperparameters in Table 1 were found empirically in preliminary experiments (if that is the case).
- In Algorithm 1, lines 9-12 contain undefined symbols. These symbols should be defined in the text to keep the paper self-contained.
- Typo on page 5 below line 152: Phase differenCE
- Figure 6: It could be interesting to show how the rhythm is changing with the velocity.
- In the discussion, the authors should compare the differences found between simulation and real deployment (if there are any).
[1] Thor, Mathias, Tomas Kulvicius, and Poramate Manoonpong. "Generic neural locomotion control framework for legged robots." IEEE transactions on neural networks and learning systems 32.9 (2020): 4013-4025.
[2] Ijspeert, Auke Jan, et al. "From swimming to walking with a salamander robot driven by a spinal cord model." science 315.5817 (2007): 1416-1420.
Author Response
Manuscript No. 2113764
Adaptive Gait Generation for Hexapod Robots Based on Reinforcement Learning and Hierarchical Framework
Responses to Reviewer#2
Dear Reviewer
Thank you for giving us an opportunity to revise our manuscript! Now we are submitting the revised manuscript entitled “Adaptive Gait Generation for Hexapod Robots Based on Reinforcement Learning and Hierarchical Framework” for consideration for publication in Sensors. Following your suggestions, we have revised and re-organized the paper carefully, all revises in manuscript we have highlighted it in blue. For details, please see the following answers.
Comment 1: I like the idea of the parametric representation of the gait combined with actor-critic policy training, which is possible to empirically evaluate with sim-to-real deployment on the hexapod robot. However, the state-of-the-art is not well represented and the claimed method properties are not experimentally corroborated.
Response: Thanks very much for your kind work and consideration on our paper. Following all your guide, we have optimized the whole paper. In experiments, we trained our framework with the common DRL algorithm to prove its generality and effectivity. Then, we trained the gait generation policy network by SAC on a simulated hexapod and verified the motion and new gaits. Besides, we compared the CoT of our gait with that of the rhythmic gait, the results show that RL-based gait has a lower CoT. Finally, we deployed the same policy on the physical VkHex robot, the comparisons between simulation and real experiments further illustrate the effectiveness of our framework.
Comment 2: Throughout the paper, the proposed RL-based method is put to contrast against CPG-based methods, which are, in my opinion, misrepresented. The dichotomy between CPG and RL-based methods (lines 55-58) is false as they can be combined into one method which adapts to the environment. Moreover, the CPG is presented as an open-loop system (lines 37-39), which is not true as CPG can synchronize with the environment and select the gaits as a response to the environment. However, rather than discussing the difference between RL-based and open-loop controllers, it would be more interesting if the authors discussed the differences between the proposed and other RL-based controllers.
Response: Thanks very much for your kind work and consideration on our paper. It is right that CPG can synchronize with the environment and select the gaits as a response to the environment. Following the above guide, we have rephrased this section and discussed the related work between the proposed and other RL-based controllers.
|
Locomotion in legged animals is characterized as a rhythmic behavior [12]. To achieve a rhythmic gait, some existing studies used Central Pattern Generators (CPG), as an open-loop oscillator [13], which rely on centrally generated rhythms to drives the overall behavior. Support for such approaches has been found in particular on fast walking in insects. Nice examples are the rapid locomotion in cockroaches [14] or running and swimming behavior of the salamander [15]. Importantly, with the high number of joints this becomes a challenging problem and open-loop oscillators solutions are usually not applicable [16]. This brought these lines of research together and has led to biologically-inspired control approaches. As one example, the Walknet approach for hexapod robots realizes a decentralized and modular structure that reflects insights from walking in stick insects [17]. While this approach can deal with a variety of disturbances during locomotion, it is still limited dealing with novel and challenging walking situations [16]. In recent years, data-driven-based methods have attracted significant attention because of their capability of making more accurate and robust control policies, which provides new feasible schemes for hexapod robot motion control [18]. The Walknet approach for hexapod robots realizes a decentralized and modular structure. Walknet can deal with a variety of disturbances during locomotion, it is still limited dealing with novel and challenging walking situations. As a type of data-driven-based method, reinforcement learning (RL) can learn feasible planning and control strategies from data and trial. RL for legged robot walking task has been extensively studied and applied. For instance, Peng et al. proposed a deep learning optimization strategy to train a biped robot in simulated animations, enabling the robot to pass through random obstacles freely [19]. In addition, Tan used proximal policy optimization to train a quadruped robot motion strategy, and realized the transformation from simulation to physical robot [20]. Similarly, Tsounis constructed the Markov decision process (MDP) and planned trajectories in the high-dimensional, continuous, state-action space, realizing the stable movement of the robot in different environments [21]. In relevant research, Deep Reinforcement Learning (DRL) has proven itself capable of automatically acquiring control policies to accomplish a large variety of challenging locomotion tasks. Fu et al. designed a novel DRL method to implement multi-contact motion planning for hexapod robots moving on uneven plum-blossom piles [22]. Specially, Thor and Manoonpong proposed a simple yet versatile modular neural control framework with fast learning, in which behavior-specific control modules can be added incrementally to obtain increasingly complex emergent locomotion behaviors [23]. Impressive performance of DRL for special tasks in footed robots. Miki employed a model-free reinforcement learning approach to train a deep policy which enables the ANYmal robot to balance a light-weight ball robustly using its limbs without any contact measurement sensor [24]. As a series-parallel composite omnidirectional mobile vehicle, the hexapod robot is a complex coupling agent. As a result, using DRL to train an end-to-end controller leads to non-convergence and gait disorder during training [25]. This has led to a growing interest in hierarchical control frameworks and how these frameworks could be exploited to improve behaviors in DRL. For example, Merel et al. proposed several such bio-inspired principles of hierarchical control and advocated their implementation also into robot architectures [26]. As one example, they emphasize the hierarchical framework with higher-level planner modulate the lower-level controller. In addition, Eppe et al. provide the cognitive foundations of hierarchical problem-solving and proposed steps to integrate biologically inspired hierarchical mechanisms to enable advanced problem-solving skills in artificial agents [27]. Both above two works provided detailed and excellent overview for hierarchical reinforcement learning, which was significantly to our paper. Similarly, Azayev and Zimmerman trained policies independently in individual scenarios using DRL, and presented a scalable two-level hierarchical for hexapod locomotion through complex terrain without the use of exteroceptive sensors [1]. |
Comment 3: Claims about the proposed algorithm are not corroborated statistically or not tested at all. In the abstract, four hypotheses are stated: proposed controller (i) is stable enough, (ii) has lower CoT than open-loop gaits, (iii) is effective, and (iv) is feasible. However, the experiments test only the second hypothesis (the proposed controller has lower CoT than open-loop gaits), and the reported results lack statistics thus I don't know whether the results are significant. In the other hypotheses (i, iii), the metrics of stability and effectivity are missing and it is not clear whether the authors compare it to the general state-of-the-art or just to the three open-loop gaits. (Minor note: The feasibility property in the fourth hypothesis might not be important as it is implied by other hypotheses.)
Response: Thanks very much for your kind comment and valuable suggestions on our paper. According to your advice, we have reorganized the comparison section. First, the algorithm in 3.F is referred to Haarnoja et al. In the experiments and results. (i) we trained our framework with the common DRL algorithm to prove its generality and effectivity. Then, (ii) we trained the gait generation policy network by SAC on a simulated hexapod and verified the motion and new gaits stability. Besides, (iii) we compared the CoT of our gait with that of the rhythmic gait, the results show that RL-based gait has a lower CoT than open-loop gaits. Finally, (iv) we deployed the same policy on the physical VkHex robot, the comparisons between simulation and real experiments further illustrate the effectiveness and feasibility of our framework.
Comment 4: The hypotheses should have clearly defined metrics and the results should show statistics of the metrics. The results should be ideally collected on the real hexapod robot.
Response: We sincerely appreciate your thoughtful suggestions and we have revised it.
Comment 5: Section 2.A. should include the description of the sensory hardware that composes the observation vector described in 2.C.
Response: Thanks for your suggestions. We have provided the relevant details and hardware of the hexapod in Table 1.
|
Table 1 VkHex details and parameters
|
Comment 6: It should be mentioned that the hyperparameters in Table 1 were found empirically in preliminary experiments (if that is the case).
Response: Special thanks to you for your good comments. In this work, the hyperparameters in Table 2 were found empirically in previous experiments. Considering your suggestion, we have revised it.
Comment 7: In Algorithm 1, lines 9-12 contain undefined symbols. These symbols should be defined in the text to keep the paper self-contained.
Response: Thanks very much for your reminder and we have revised it and checked all symbols. In Algorithm 1, , and are the gradients and are the approximate gradient functions.
Comment 8: Typo on page 5 below line 152: Phase differenCE
Response: Thanks for pointing out my mistake, we are very sorry for our incorrect writing. Meanwhile, we have carefully checked the formulations of the whole paper and revised it.
Comment 9: Figure 6: It could be interesting to show how the rhythm is changing with the velocity.
Response: Thanks very much for your consideration in Figure 6. In the simulation and real experiment, the hexapod robot gradually transitions from a quadrangular-like gait to a triangular-like gait as the speed increases. To show how the rhythm is changing with the velocity, we have detailed description for Figure 9 and Figure 10. we would like to express our sincere gratitude again for this constructive comment.
|
Figure 9. Speed-adaptive gaits of hexapod robot trained in the hierarchical framework for reinforcement learning. From (a) to (h), the robot walk in a fixed direction and the motion speed gradually increases to 0.6m/s. As the speed increases, the hexapod robot gradually transitions from a wave -like gait to a tripod-like gait, and the robot is stable enough not to fall. Figure 10. Visualization of legs’ phase for the RL-based gaits. Black rectangles indicate support phases, gray rectangles indicate swing phases, and block lengths indicate duration. The hexapod robot walks with a wave-like gait in the first cycle, a quadruped-like gait in the second to fourth cycles, and finally walks in a tripod-like gait. Within each gait cycle, RL-based gait has a different duty cycle and phase difference. |
Comment 10: In the discussion, the authors should compare the differences found between simulation and real deployment (if there are any).
Response: Thanks very much for your kind and valuable consideration. Considering your suggestions, we have revised this part and discussed the differences between simulation and real.
|
The previous simulations illustrate that gait policy network can trained by our framework and is optimized enough to perform adaptive and stable gaits. In this experiment, we deployed the same policy on the physical VkHex robot and the policy ran 100 Hz on VkHex. As the movement speed increased from 0m/s to 0.5m/s, the RL-based gaits of the hexapod robot behaved as shown in Figure 11. Compared with existing rhythmic gait, the RL-based gaits perform more flexibility. As speed increased, VkHex could adaptively generate rhythmic-like gaits. Furthermore, all gaits were stable during locomotion which performed in the real world was familiar to the simulated RL-based gaits. Since the environment disturbances and modeling uncertainties between simulation and reality are different, such as servo force, friction coefficient, sensor noise and inaccurate motor models, the variety and stability of the gait in real is slightly poor than in simulation. The results of simulation and real experiments further illustrate the effectiveness of the proposed framework. Figure 11. RL-based hierarchical framework tested on the VkHex prototype on flat terrain. Similar from (a) to (h), the robot walked in a fixed direction on while the motion speed gradually increased to 0.5m/s. The hexapod robot walked primarily in a triangular-like gait and gradually to the left. |
Once again, sincere thanks for your time and effort in further processing our revised paper. Looking forward to your guidance again.

Reviewer 3 Report
1.The research results can be detailed in the abstract with more sentences to help readers understanding main contributions quickly.
2.You should describe the relation of your research with actuators in the main text of the manuscript.
3.You should state what’s meaning for the “a robust adaptive control framework” in the abstract. Do you include the influences of environment disturbances and modeling uncertainties in this research? The hexapod robot only tested with flat terrain and no any loading.
4. All of equations number should be right-justified.
5.You should describe each component for more details in Figure 2.
For example: what are the “Trajectory Tracking Controller” …..?
6.There are no detailed descriptions for each figure. You should state what meaning is for each figure and more details. This is very important to help readers understanding your key contributions, research structures and results.
7.You should indicate what the robustness under environment disturbances and modeling uncertainties and how to eliminate these influences in your research.
8.You should test this hexapod robot around rough terrain and attach to some loading.
9.You should provide the hexapod robot system parameters, initial position, attitude angle in your simulation scenarios.
10.You should compare with more researches to clarify the optimization of this proposed research.
11.You should describe what is the proposed algorithm 1 for more details in this manuscript. This is very important to help readers understanding what is your key operation logic.
12.You should provide more explanations of the conclusion from corresponding studies.
Author Response
Manuscript No. 2113764
Adaptive Gait Generation for Hexapod Robots Based on Reinforcement Learning and Hierarchical Framework
Responses to Reviewer#3
Comment 1: The research results can be detailed in the abstract with more sentences to help readers understanding main contributions quickly.
Response: Thanks very much for your kind work and consideration on our paper. Following the above guide, we have removed the overview of the article and detailed experimental results and main contributions. The improved abstract is displayed below.
|
In general, the hexapod robot walks with a fixed rhythmic gait, which reduces its adaptability and limits motion efficiency. Moreover, the hexapod robot has a high-dimensional action space, it has a great challenge to use reinforcement learning to directly train the robot's joint angles. As a result, a hierarchical and modular framework and learning details are proposed in this paper, using only seven-dimensional vectors to denote the agent actions. In addition, we conduct experiments and deploy the proposed framework to a real hexapod robot. The experimental results show that superior reinforcement learning algorithms can converge in our framework, such as SAC, PPO, DDPG and TD3. Specifically, the gait policy trained in our framework can generate new adaptive hexapod gait on flat terrain, which is stable enough and has lower transportation cost than rhythmic gaits. |
Comment 2: You should describe the relation of your research with actuators in the main text of the manuscript.
Response: Thank you for your helpful suggestions. We have made corrections and rephrased this section to address your suggestions.
|
This paper studies the gait generation of hexapod robots and realizes adaptive gait generation using RL and hierarchical framework. The main contributions of this paper are as follows. First, we develop a RL-based hierarchical control framework to reduce the dimensionality of the action space, which is then deployed in a real robot. Specifically, for the speed-adaptive gait generation task, we describe the whole learning process, including the MDP formulation, detailed training settings, and policy gradient algorithm. Furthermore, we design simulations and experiments to demonstrate our framework’s effectiveness. Overall, the proposed hierarchical framework appears novel and unique enough to be applied in reinforcement learning. In addition, we designed specific trajectory planner, inverse kinematics solver and trajectory tracking controller for hexapod robot. This paper makes a valuable contribution to the actuators journal. |
Comment 3: You should state what’s meaning for the “a robust adaptive control framework” in the abstract. Do you include the influences of environment disturbances and modeling uncertainties in this research? The hexapod robot only tested with flat terrain and no any loading.
Response: Special thanks to you for your rigorous advice, and I’m sorry for my uncritical presentation. In our research, we adopt some details to improve the robustness, including randomizing model parameters, introducing sensor noise and applying external disturbance. We have rewritten the abstract as follows.
|
In general, the hexapod robot walks with a fixed rhythmic gait, which reduces its adaptability and limits motion efficiency. Moreover, the hexapod robot has a high-dimensional action space, it has a great challenge to use reinforcement learning to directly train the robot's joint angles. As a result, a hierarchical and modular framework and learning details are proposed in this paper, using only seven-dimensional vectors to denote the agent actions. In addition, we conduct experiments and deploy the proposed framework to a real hexapod robot. The experimental results show that superior reinforcement learning algorithms can converge in our framework, such as SAC, PPO, DDPG and TD3. Specifically, the gait policy trained in our framework can generate new adaptive hexapod gait on flat terrain, which is stable enough and has lower transportation cost than rhythmic gaits. |
Comment 4: All of equations number should be right-justified.
Response: Thank you very much for your suggestions, we have revised it.
Comment 5: You should describe each component for more details in Figure 2. For example: what are the “Trajectory Tracking Controller” ?
Response: Thanks for your valuable and insightful suggestions. Following your suggestions, have described the whole framework in Section 2.B.a) ~ 2.B.e), including gait modulator, trajectory planner, IK solver and tracking controller.
|
e)Trajectory Tracking Controller In hierarchical framework, we adopt a sliding mode controller as the joint trajectory tracking controller and use a nonlinear potential-like function to place the integral:
where is the regulator and is the joint position error. The control law of the nonlinear integral sliding mode control can be expressed as:
where is the pseudo-inverse matrix of the linear velocity Jacobi matrix, is the sliding surface, and are estimates for adaptation laws, is the ideal foot-end velocity trajectory. Figure 7 shows the system block diagram of the nonlinear integral sliding mode controller.
|
Comment 6: There are no detailed descriptions for each figure. You should state what meaning is for each figure and more details. This is very important to help readers understanding your key contributions, research structures and results.
Response: Thank you for your helpful guide. Based on the above suggestion, we have stated detailed description for each figure.
Comment 7: You should indicate what the robustness under environment disturbances and modeling uncertainties and how to eliminate these influences in your research.
Response: Thanks for your valuable and insightful suggestions. Following your guide, we have revised it. Environment bias and modeling uncertainties between the simulation and physical robot affects the porting of the RL model. In simulation, we adopt some commonly used details to improve the robustness.
Comment 8: You should test this hexapod robot around rough terrain and attach to some loading.
Response: Special thanks to you for your good comments. In this paper, we aim to design a RL-based hierarchical framework for multi-joint hexapod, using extremely simple observations and actions to implement adaptive gait generation. Consider your advice, we have applied random external force to the hexapod robot during training, like loading. Our further research focus on the differences in reward and observation design under our RL-based framework for challenging terrains and loading tasks.
Comment 9: You should provide the hexapod robot system parameters, initial position, attitude angle in your simulation scenarios.
Response: Thank you very much for your suggestion. We have provided the hexapod robot system parameters, initial position, attitude angle in Table 1 and Section 4.C
|
Table 1 VkHex details and parameters
|
Section 4.C
|
The initial height of the robot is 0.1m and the direction is the y-axis of the body coordinate frame. We train our robot by giving a specific target in 1m front. Once it reaches its initial target, the next goal is given again in 1m ahead repeatedly. We transport the gait policy network to the simulation robot, gradually increase the expected velocity to 0.6m/s. Figure 9 shows the motion process of the hexapod robot in 7 continuous motion cycles, and Figure 10 shows the corresponding gait phase. |
Comment 10: You should compare with more researches to clarify the optimization of this proposed research.
Response: Thank you for your helpful analysis and suggestions. We have compared with more related studies and rephrased this section to address your suggestions. To draw out our approach, we have introduced the CPG and Walknet approach and work to achieve a rhythmic behavior, and added related work for reinforcement learning approaches in robot control. In addition, we used a new section to introduce some work on hierarchical DRL for robot locomotion. The modified comparison is shown as below:
|
Locomotion in legged animals is characterized as a rhythmic behavior [12]. To achieve a rhythmic gait, some existing studies used Central Pattern Generators (CPG), as an open-loop oscillator [13], which rely on centrally generated rhythms to drives the overall behavior. Support for such approaches has been found in particular on fast walking in insects. Nice examples are the rapid locomotion in cockroaches [14] or running and swimming behavior of the salamander [15]. Importantly, with the high number of joints this becomes a challenging problem and open-loop oscillators solutions are usually not applicable [16]. This brought these lines of research together and has led to biologically-inspired control approaches. As one example, the Walknet approach for hexapod robots realizes a decentralized and modular structure that reflects insights from walking in stick insects [17]. While this approach can deal with a variety of disturbances during locomotion, it is still limited dealing with novel and challenging walking situations [16]. In recent years, data-driven-based methods have attracted significant attention because of their capability of making more accurate and robust control policies, which provides new feasible schemes for hexapod robot motion control [18]. The Walknet approach for hexapod robots realizes a decentralized and modular structure. Walknet can deal with a variety of disturbances during locomotion, it is still limited dealing with novel and challenging walking situations. As a type of data-driven-based method, reinforcement learning (RL) can learn feasible planning and control strategies from data and trial. RL for legged robot walking task has been extensively studied and applied. For instance, Peng et al. proposed a deep learning optimization strategy to train a biped robot in simulated animations, enabling the robot to pass through random obstacles freely [19]. In addition, Tan used proximal policy optimization to train a quadruped robot motion strategy, and realized the transformation from simulation to physical robot [20]. Similarly, Tsounis constructed the Markov decision process (MDP) and planned trajectories in the high-dimensional, continuous, state-action space, realizing the stable movement of the robot in different environments [21]. In relevant research, Deep Reinforcement Learning (DRL) has proven itself capable of automatically acquiring control policies to accomplish a large variety of challenging locomotion tasks. Fu et al. designed a novel DRL method to implement multi-contact motion planning for hexapod robots moving on uneven plum-blossom piles [22]. Specially, Thor and Manoonpong proposed a simple yet versatile modular neural control framework with fast learning, in which behavior-specific control modules can be added incrementally to obtain increasingly complex emergent locomotion behaviors [23]. Impressive performance of DRL for special tasks in footed robots. Miki employed a model-free reinforcement learning approach to train a deep policy which enables the ANYmal robot to balance a light-weight ball robustly using its limbs without any contact measurement sensor [24]. As a series-parallel composite omnidirectional mobile vehicle, the hexapod robot is a complex coupling agent. As a result, using DRL to train an end-to-end controller leads to non-convergence and gait disorder during training [25]. This has led to a growing interest in hierarchical control frameworks and how these frameworks could be exploited to improve behaviors in DRL. For example, Merel et al. proposed several such bio-inspired principles of hierarchical control and advocated their implementation also into robot architectures [26]. As one example, they emphasize the hierarchical framework with higher-level planner modulate the lower-level controller. In addition, Eppe et al. provide the cognitive foundations of hierarchical problem-solving and proposed steps to integrate biologically inspired hierarchical mechanisms to enable advanced problem-solving skills in artificial agents [27]. Both above two works provided detailed and excellent overview for hierarchical reinforcement learning, which was significantly to our paper. Similarly, Azayev and Zimmerman trained policies independently in individual scenarios using DRL, and presented a scalable two-level hierarchical for hexapod locomotion through complex terrain without the use of exteroceptive sensors [1]. |
Comment 11: You should describe what is the proposed algorithm 1 for more details in this manuscript. This is very important to help readers understanding what is your key operation logic.
Response: Thank you very much for your reminder, we have revised it and described more details about SAC. SAC is an off-policy maximum entropy DRL algorithm where the actor aims to maximize expected reward and entropy. In this framework, SAC provides sample-efficient learning while retaining the benefits of entropy maximization and stability.
Comment 12: You should provide more explanations of the conclusion from corresponding studies.
Response: We sincerely appreciate your thoughtful suggestions and we have revised this section as follows.
|
In this paper, a RL-based hierarchical framework is proposed to simplify the hexapod action and learn a RL-based gait generation task, including policy network, gait planner, IK solver and trajectory tracking controller. Furthermore, to train an adaptive gait generation policy network, we designed a whole RL framework for the hexapod robot and tested our framework using SAC, PPO, DDPG and TD3 in a physics simulation. Our framework enabled the DRL algorithms to converge and allowed the hexapod to generate adaptive new gaits on flat terrain. Finally, we run the same policy in the real robot without any modification. Through comparison and experiments, it is verified that this framework is effective for RL task. This paper realizes the speed-adaptive gait generation and planning of the hexapod robot. We only tested with flat terrain, so further research can include a study of the similarities and differences in reward design under the RL framework for challenging terrains and tasks. |
Comment 13: Line 265 and 269: I found these statements too strong. They are not backed up by the authors table and they don’t show other controllers.
Response: Thanks for your suggestions. I’m sorry for my unreasonable presentation, what we to compare is the flexibility performance between the existing rhythmic gait and RL-based gait. we have revised this section as follows.
|
The previous simulations illustrate that gait policy network can trained by our framework and is optimized enough to perform adaptive and stable gaits. In this experiment, we deployed the same policy on the physical VkHex robot and the policy ran 100 Hz on VkHex. As the movement speed increased from 0m/s to 0.5m/s, the RL-based gaits of the hexapod robot behaved as shown in Figure 11. Compared with existing rhythmic gait, the RL-based gaits perform more flexibility. As speed increased, VkHex could adaptively generate rhythmic-like gaits. Furthermore, all gaits were stable during locomotion which performed in the real world was familiar to the simulated RL-based gaits. Since the environment disturbances and modeling uncertainties between simulation and reality are different, such as servo force, friction coefficient, sensor noise and inaccurate motor models, the variety and stability of the gait in real is slightly poor than in simulation. The results of simulation and real experiments further illustrate the effectiveness of the proposed framework. Figure 11. RL-based hierarchical framework tested on the VkHex prototype on flat terrain. Similar from (a) to (h), the robot walked in a fixed direction on while the motion speed gradually increased to 0.5m/s. The hexapod robot walked primarily in a triangular-like gait and gradually to the left. |
Once again, sincere thanks for your time and effort in further processing our revised paper. Looking forward to your guidance again.

Round 2
Reviewer 2 Report
I commend the authors for adding more detail to the methodology description, which greatly increased the reproducibility of the proposed method. Still, however; the experimental section lacks a report on the repeatability of the results, which is an essential part of the scientific paper.
Major issue
As it is now, the Experiments section reports only one instance of the experimental setup. Authors can show the results' repeatability simply by repeating the same experiment multiple times (e.g., five times) and showing the result statistics, for example, by Figure 8 and Table 4 showing mean and standard deviation.
Especially in section 4E, which reports on real deployment, I would expect a report on repeated experiments with more measured metrics. First, authors should add measured metrics during the real deployment. For example, compare the output of the IMU measured during "rhythmic gaits" and the proposed controller. Second, the authors should report on the statistics of the repeated experiment. For example, the mean value of the IMU output with standard deviations. Such data would then corroborate claims authors made in lines 319-322. Other metrics might be CoT or the difference between set velocity and measured velocity.
Minor issues and recommendations
- I find the first (line 4) sentence of the abstract misleading: "In general, the hexapod robot walks with fixed rhythmic gait," this is a strong claim that should be grounded by a review on hexapod control, where the majority of cited research uses fixed rhythmic gait. However, it is true that: "If the hexapod robot walks with fixed rhythmic gaits, it reduces its adaptability." Thus I think it is enough to reformulate the sentence on line 4 in order to make the statement true.
- In my opinion, mentioning the "data-driven methods" after two long paragraphs is late. Authors might want to inform the reader about the focus on "gait control" and "data-driven methods" sooner.
- Table 1: The "Developer" should be "Computing device".
- Equation (7): I think the nominator and denominator are switched (the fraction term is not defined for x=0, while the tanh is).
- Line 211: How is the velocity measured/estimated?
- Table 4: I think these data points could be shown by a plot with velocity on X-axis and CoT on Y-axis.
- Figure 7: There should be an indication of duration between the frames. Also indicating the scale (like in cartographic maps) would improve the presentation.
Author Response
Dear Reviewer
Thanks again for your valuable and insightful suggestions on my revised paper. Following your suggestions, we have revised and re-organized the paper again, all revises in manuscript we have highlighted it in blue. For details, please see the following answers.
Comment 1: As it is now, the Experiments section reports only one instance of the experimental setup. Authors can show the results' repeatability simply by repeating the c multiple times (e.g., five times) and showing the result statistics, for example, by Figure 8 and Table 4 showing mean and standard deviation.
Response: Thanks very much for your detailed suggestion. Referring to your suggestion, we have repeated experiment 4D three times and show the mean transportation cost.
Comment 2: Especially in section 4E, which reports on real deployment, I would expect a report on repeated experiments with more measured metrics. First, authors should add measured metrics during the real deployment. For example, compare the output of the IMU measured during "rhythmic gaits" and the proposed controller. Second, the authors should report on the statistics of the repeated experiment. For example, the mean value of the IMU output with standard deviations. Such data would then corroborate claims authors made in lines 319-322. Other metrics might be CoT or the difference between set velocity and measured velocity.
Response: Special thanks for pointing out the shortcomings of section 4E and for your detailed and wise suggestions. Since the COVID-19 and winter vacation, we are sorry that the hexapod robot is not around and we are temporarily unable to repeat experiments with more measured metrics.
Comment 3: I find the first (line 4) sentence of the abstract misleading: "In general, the hexapod robot walks with fixed rhythmic gait," this is a strong claim that should be grounded by a review on hexapod control, where the majority of cited research uses fixed rhythmic gait. However, it is true that: "If the hexapod robot walks with fixed rhythmic gaits, it reduces its adaptability." Thus I think it is enough to reformulate the sentence on line 4 in order to make the statement true.
Response: Thanks for your wise consideration and suggestion, and we have revised and reformulated line 4 to avoid misleading.
|
Gaits play a decisive role to the performance of hexapod robot walking, this paper focus on the adaptive gait generation with reinforcement learning for a hexapod robot. |
Comment 4: In my opinion, mentioning the "data-driven methods" after two long paragraphs is late. Authors might want to inform the reader about the focus on "gait control" and "data-driven methods" sooner.
Response: Thanks for your insightful suggestions. Following your suggestions, we have optimized this section. In introduction, we list the advantages and challenges in hexapod robot gait and motion in paragraph 1. Then, we show the challenges of two existing studies in hexapod robots, including CPG and Walknet, and inform the reader this paper use the data-driven method to complete hexapod gait generation.
Comment 5: Table 1: The "Developer" should be "Computing device".
Response: Thanks very much for your recommendation and we have replaced them in Table 1.
Comment 6: Equation (7): I think the nominator and denominator are switched (the fraction term is not defined for x=0, while the tanh is).
Response: Thanks for pointing out my mistake. We are sorry for my carelessness that the numerator and denominator of Equation (7) are reversed and we have switched them.
Comment 7: Line 211: How is the velocity measured/estimated?
Response: Thanks very much for your question. In Pybullet physics simulation engine, we can use getBaseVelocity() to measure the base velocity. Since the jitter of the fuselage during the movement, obtaining the speed by integrating the acceleration of the IMU has a large cumulative error. In VkHex, we use an optical flow-based speed sensor to measure the base velocity.
Comment 8: Table 4: I think these data points could be shown by a plot with velocity on X-axis and CoT on Y-axis.
Response: Special thanks for your helpful suggestions, we have shown the data by a Figure.
Comment 9: Figure 7: There should be an indication of duration between the frames. Also indicating the scale (like in cartographic maps) would improve the presentation.
Response: Special thanks for your helpful suggestions. Following your suggestions, we have indicated the duration between the frames and the scale to improve the presentation.
Once again, sincere thanks for your time and effort in further processing our revised paper. Looking forward to your guidance again.

Reviewer 3 Report
Thanks for your response.
For “Comment 4: All of equation numbers should be right-justified.”, some equation numbers still did not right-justified and there are two equations (11) in your manuscript.
Please check this issue and revise it.
Author Response
Dear Reviewer
Thanks again for your valuable and insightful suggestions on my revised paper. Following your suggestions, we have revised and re-organized the paper again, all revises in manuscript we have highlighted it in blue. For details, please see the following answers.
Comment 1: For “Comment 4: All of equation numbers should be right-justified.”, some equation numbers still did not right-justified and there are two equations (11) in your manuscript.
Response: Special thanks to you for your good reminders. We have rewritten all equations to be right-justified instead of hidden table.
Original form:
|
(11) |
Modified form:
(11)
Once again, sincere thanks for your time and effort in further processing our revised paper. Looking forward to your guidance again.

Round 3
Reviewer 2 Report
I believe that the journal content improved. It is a pity that the deployment experiments can't be repeated.
I have then some minor comments:
1) Fig 11: The mean are visualised, but the standard deviations should be indicated as well.
2) Fig 12: The images have too low resolution, I can't read the newly added text.
Author Response
Dear Reviewer
Thanks again for your valuable and insightful suggestions on my revised paper. Following your suggestions, we have revised and re-organized the paper again, all revises in manuscript we have highlighted it in blue.
Comment 1: Fig 11: The mean are visualised, but the standard deviations should be indicated as well.
Response: Special thanks for your wise consideration. According to your consideration we have indicated the standard deviations in Fig 11 b).
Comment 2: Fig 12: The images have too low resolution, I can't read the newly added text.
Response: Thanks very much for your suggestion, we have enlarged the text and improved the resolution in Fig 12.
Once again, sincere thanks for your time and effort in further processing our revised paper. Looking forward to your guidance again. Today is Chinese New Year, the Spring Festival, warm hearted wishes for a happy New Year filled with all your favorite.
